# Tumour-resident oncolytic bacteria trigger potent anticancer effects through selective intratumoural thrombosis and necrosis

Seigo Iwata[1], Taisei Nishiyama[1], Matomo Sakari[1], Yuki Doi[2], Naoki Takaya[2], Yusuke Ogitani[3], Hiroshi Nagano[3], Keisuke Fukuchi[3] & Eijiro Miyako [1]✉

Intratumoural bacteria represent a promising drug-free strategy in cancer therapy. Here we demonstrate that a tumour-resident bacterial consortium—*Proteus mirabilis* (A-gyo) and *Rhodopseudomonas palustris* (UN-gyo)—in a precise 3:97 ratio (A-gyo:UN-gyo), exhibits potent antitumour efficacy independent of immune cell infiltration. In both immunocompetent and immunocompromised mouse models, including human tumour xenografts, intravenous administration of the bacterial consortium led to complete tumour remission, prolonged survival, and no observable systemic toxicity or cytokine release syndrome. Genomic and phenotypic analyses revealed A-gyo's unique non-pathogenic profile and impaired motility, while UN-gyo modulated A-gyo's biogenic activity, enhanced safety and promoted cancer-specific transformation. Mechanistically, the bacterial consortium triggered selective intratumoural thrombosis and vascular collapse—supported by cytokine induction, fibrin deposition and platelet aggregation—culminating in widespread tumour necrosis. The consortium also proliferated within tumours, formed biofilms and exerted direct oncolytic effects. This natural bacterial synergy—achieved without genetic engineering—offers a self-regulating and controllable strategy for safe, tumour-targeted therapy.

The advent of cancer immunotherapies using checkpoint inhibitors[1,2] and chimeric antigen receptor (CAR) T cells[3,4] has revolutionized the oncology paradigm, positioning these therapies as a fourth pillar for patients alongside surgery, chemotherapy and radiation. However, their application in solid tumours has been restricted by their limited ability to penetrate and function in the immunosuppressive tumour microenvironment, especially within immune-privileged hypoxic cores[5–7]. In fact, patients with cancer undergoing active chemotherapy and/or radiation treatments are typically immunosuppressed[8,9]. Given such intravital immunosuppressive conditions, the low therapeutic efficacy of immunotherapy in cancer treatment remains a major challenge.

Recent advances in microbial biotechnology and genetic engineering might overcome the drawbacks of current cancer immunotherapy associated with immunological dysfunction by designing genomic circuits to secrete antitumour agents that directly destroy tumours[10–15]. This therapeutic strategy uses anaerobic bacteria to specifically target and colonize the hypoxic tumour tissues with preferential accumulation and proliferation[16]. Engineered bacteria can be manipulated to directly deliver therapeutic payloads such as cytotoxic agents, immunomodulators, cytokines, prodrug-converting enzymes, small interfering RNAs and nanobodies into the tumour microenvironment at effective dosages[17]. Once colonized in the tumour, bacteria actively

[1]Graduate School of Advanced Science and Technology, Japan Advanced Institute of Science and Technology, Nomi, Japan. [2]Faculty of Life and Environmental Sciences, Microbiology Research Center for Sustainability, University of Tsukuba, Tsukuba, Japan. [3]Discovery Research Laboratories I, Research Function, R&D Division, Daiichi Sankyo Co., Ltd, Shinagawa, Japan. ✉e-mail: e-miyako@jaist.ac.jp

proliferate within the immune-privileged tumour hypoxia to induce antitumour efficacy by increasing immune surveillance and decreasing immunosuppression[18]. Since the notion of treating solid tumours with living bacteria was reported as a progenitor of immunotherapy more than 150 years ago, bacteria-based cancer therapies fundamentally rely on cytotoxic immune cells in the tumour milieu to ensure substantial anticancer drug efficacy[19]. However, despite the complex genetic procedures involved in engineering highly pathogenic bacteria, such as *Salmonella typhimurium* and *Listeria monocytogenes*, for this therapeutic strategy, engineered bacteria cannot fully eliminate tumours[17].

Intratumoural bacteria, first identified inside tumours in the late nineteenth century, could play a key role as a modality for modern cancer therapy[20–26]. Previously, we discovered highly targeted cancer therapeutic non-pathogenic bacteria, named AUN, in the tumour-resident microbiota and confirmed their association with natural photosynthetic bacteria[27]. We demonstrated that the intratumoural bacterial consortium AUN, composed of *Proteus mirabilis* (A-gyo) and *Rhodopseudomonas palustris* (UN-gyo), showed dramatic anticancer responses in various syngeneic mouse models, including colorectal cancer, sarcoma, metastatic lung cancer and extensive drug-resistant triple-negative breast cancer. However, the reason behind the anticancer efficacy and AUN biocompatibility remains unclear. In addition, its anticancer effects and mechanisms of action in immunocompromised models remain unknown.

This study explored the unique microbial characteristics and potential antitumour efficacy of the AUN bacterial consortium in various immunocompromised mouse models. Furthermore, we assessed AUN anticancer efficacy in mouse and human cancer-bearing immunocompromised mice, including tumour suppression and biocompatibility, and analysed local and systemic immunological responses and cellular behaviours in the blood, spleen and tumour tissues to gain mechanistic insights into its antitumour effect and biocompatibility. We also found that UN-gyo could enhance anticancer efficacy and AUN safety via the following functions: (i) suppression of biogenic activity (pathogenicity) of A-gyo, (ii) increase of cancer-specific cytotoxicity through the facilitation of fibrous structural transformation of A-gyo in the presence of cancer cells (oncometabolites), (iii) attenuation of the haemolytic activity of AUN and (iv) increase of iron requirements of AUN. This study provides an effective approach for investigating a unique therapeutic strategy for the treatment of immunocompromised patients with cancer.

## Results

### In vitro analysis of microbial features of AUN

To unveil the genomic characteristics of the AUN bacterial consortium, comparative full-genome analyses of A-gyo and UN-gyo were performed and compared with the commercially available *P. mirabilis* (ca-PM) and *R. palustris* (ca-RP) strains. Surprisingly, A-gyo was completely defective in pathogenic gene factors, such as pili and adhesin (Fig. 1a and Supplementary Data 1). We believe that this is why the A-gyo bacteria could reside in living mice. In our previous study, mice injected with ca-RP presented movement disorders (crouching position and shivering) and hypothermia[27]. By contrast, the genomic features of UN-gyo fully matched those of ca-RP because UN-gyo was originally isolated from a solid tumour after intravenous (i.v.) ca-RP administration[27]. This result indicates that the UN-gyo genome function is intrinsically the same as ca-RP, and mutation of ca-RP did not occur when the bacteria were physically isolated from the tumour after i.v. injection.

Transmission electron microscopy also confirmed that A-gyo was defective in the fibrous pili on its surface, although a few long flagella were also observed (Fig. 1b). It is known that ca-PM can migrate across solid media surfaces using a type of cooperative group motility called swarming[28,29]. Although conventional ca-PM showed a characteristic wave pattern owing to swarming on the agar plate, the tumour-resident A-gyo could not form this pattern because of the lack of pili and adhesin

genetic factors (Fig. 1c). Similarly, the slower A-gyo movement compared with ca-PM found in the in vitro motility assay was due to the genetic absence of pili motor function in A-gyo (Fig. 1d).

Maintaining precise bacterial ratios is important for the translational and clinical feasibility of tumour-resident bacterial consortia. Even after repeated subculturing, the AUN ratio was consistently A-gyo:UN-gyo ≈ 3:97, as confirmed using fluorescence microscopy and colony assays[27]. We have further analysed the bacterial ratios in media using quantitative polymerase chain reaction (qPCR) (Supplementary Figs. 1 and 2). As a result, the AUN ratio is A-gyo:UN-gyo ≈ 3:97 even after repeated subculturing (Supplementary Fig. 1). We also confirmed that the ratios naturally returned to the original AUN ratio (A-gyo:UN-gyo ≈ 3:97) 7 days after culturing, even when the ratios were intentionally prepared as A-gyo:UN-gyo = 10:90, 50:50 or 97:3 in advance (Supplementary Fig. 2). These results may indicate that A-gyo has a symbiotic relationship (commensalism) with UN-gyo owing to the Cys metabolism in A-gyo cells, as we previously reported[27]. Meanwhile, the artificial mixed AUN ratios (A-gyo:UN-gyo = 15:85, 25:75, 50:50 or 97:3) caused Colon26-bearing BALB/c mice (*N* = 5 for each ratio) to die within 48 h. Therefore, we believe that nature-prepared 'golden' AUN ratio (A-gyo:UN-gyo ≈ 3:97) has strong anticancer efficacy and safety. Interestingly, in the transcriptome analyses, the AUN bacterial consortium exhibited a completely different gene expression pattern from that of the purified single strain UN-gyo, even though the bacterial majority within the AUN was UN-gyo (Fig. 1e). In addition, the biogenic activity of A-gyo was notably suppressed by the presence of UN-gyo in AUN rather than that of the mono-cultured A-gyo (Supplementary Data 2). Surprisingly, compared with UN-gyo, AUN specifically increased gene expression related to extracellular iron acquisition for siderophore and haem metabolism, likely owing to bacterial cross-talk, such as biochemical reactions and interbacterial signalling[30,31] (Fig. 1f, Supplementary Table 1 and Supplementary Data 2). Cancerous tumours typically use iron for cancer initiation, tumour growth and metastasis[32]. We speculate that AUN may also deplete iron from cancerous tumours and blood in the tumour microenvironment, leading to tumour-specific suppression (a mechanism described in detail in the 'Mechanism of tumour suppression')[33,34]. Therefore, we consider that the increased iron requirements of AUN could be a key factor for potent anticancer regression. The final microbial feature of interest was that the viability of the potent anticancer therapeutic bacteria AUN, comprising A-gyo and UN-gyo, could be controlled by antibiotics (Fig. 1g and Supplementary Fig. 3). AUN was eliminated from the mice by antibiotic imipenem administration (Supplementary Fig. 4). Despite the high controllability of AUN using antibiotics, concerns such as potential drug resistance and infection risks associated with the use of live bacteria require careful consideration and resolution. The preliminary findings of this research, although promising, need further exploration to fully comprehend and validate the potential risks associated with toxicity and efficacy.

### Antitumour efficacy of AUN in various immunocompromised models

At the beginning of the study, we did not expect AUN to have sufficient antitumour efficacy in an immunocompromised model without a systemic healthy immune system, as we believed that, similar to other bacteria-based cancer therapies, cytotoxic immune cells play a major role in achieving the antitumour efficacy of AUN. To explore the unexpected efficacy of AUN in immunocompromised models, mouse Colon26 tumour-bearing BALB/c-nu/nu mice were intravenously injected with a single or double dose of AUN (Fig. 2a). This model was chosen because thymus-deficient BALB/c-nu/nu mice lack T lymphocytes[35]. Against all expectations, AUN showed dramatic antitumour efficacy similar to the previously reported immunocompetent mice[27] just by a single AUN administration ($0.5 \times 10^8$ CFU ml$^{-1}$, $2.5 \times 10^8$ CFU ml$^{-1}$, $5 \times 10^8$ CFU ml$^{-1}$, $1 \times 10^9$ CFU ml$^{-1}$, $3.0 \times 10^9$ CFU ml$^{-1}$, $5.0 \times 10^9$ CFU ml$^{-1}$ or

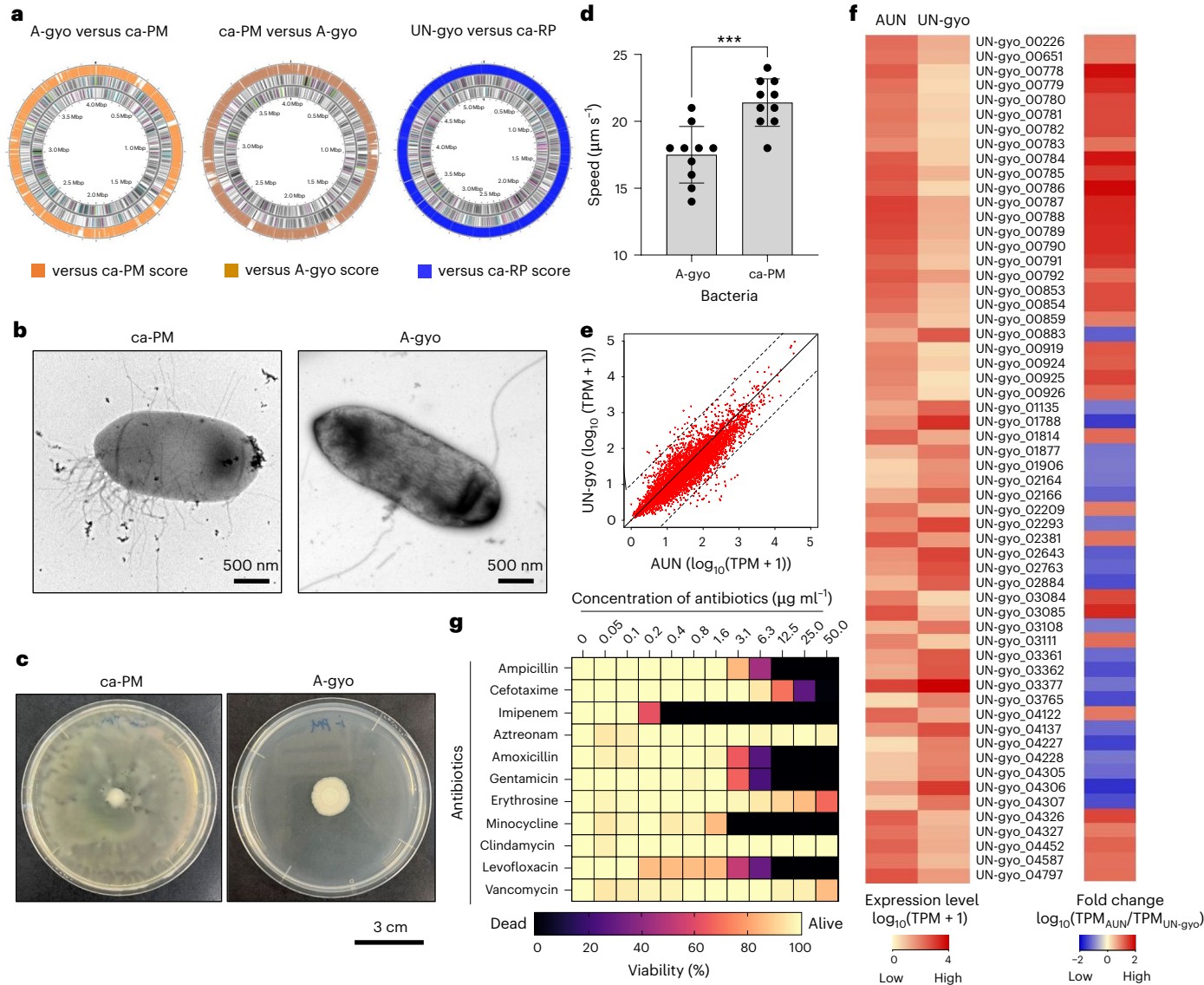

**Fig. 1 | Characterization of the bacterial consortium AUN. a**, Comparative genomic analyses of A-gyo versus *P. mirabilis* (ca-PM) (left), ca-PM versus A-gyo (centre) and UN-gyo versus *R. palustris* (ca-RP) (right). **b**, Transmission electron microscopy images of ca-PM (left) (*n* = 3 independent experiments) and A-gyo (right) (*n* = 3 independent experiments). **c**, Photos of bacterial colonies of ca-PM (left) and A-gyo (right). **d**, Bacterial motility assay. Data are represented as the mean ± s.e.m.; *n* = 10 independent experiments. ***P = 0.001 by Student's two-sided *t*-test. **e**, The statistics of differential expression genes of AUN and UN-gyo, based on the $\log_{10}$ of transcripts per million (TPM). The solid black line represents the line of equality (*y* = *x*), indicating identical expression levels between AUN and UN-gyo, while the dashed lines show boundaries for deviations from equality, representing fold-change thresholds. **f**, Heat map for the gene expression level of AUN and UN-gyo (left) and comparative gene expression of AUN/UN-gyo (right) based on the $\log_{10}$ of TPM. **g**, Minimum inhibitory concentration test of various antibiotics against AUN. Data are represented as mean ± s.e.m.; *n* = 3 independent experiments.

$7.8 \times 10^9$ CFU ml$^{-1}$) or from double dose two of the bacterial suspension using a combination of low and high AUN doses ($1 \times 10^7$ CFU ml$^{-1}$ and $15 \times 10^9$ CFU ml$^{-1}$) (Fig. 2b,c and Supplementary Fig. 5). The tumours darkened within 24 h of a single i.v. injection and after the second injection of double-dose AUN. The colour change was likely due to tumour-specific thrombosis (a mechanism described in detail in the section 'Mechanism of tumour suppression'). Unfortunately, despite experiencing major initial tumour suppression, the group receiving a single AUN dose ($7.8 \times 10^9$ CFU ml$^{-1}$) demonstrated tumour reoccurrence on day 13. The control group administered PBS (placebo) did not show any tumour regression. The maximum tolerated dose of a single AUN administration was $7.8 \times 10^9$ CFU ml$^{-1}$, as doses higher than this caused mice to die within 1 or 2 days after i.v. administration (Supplementary Table 2). Coincidentally, we found that the fatal concentration of AUN ($>7.8 \times 10^9$ CFU ml$^{-1}$) as a single dose was not fatal if injected

2 days after an i.v. injection of a low dose of AUN ($1 \times 10^7$ CFU ml$^{-1}$). Interestingly, the double dose of AUN (1st dose, $1 \times 10^7$ CFU ml$^{-1}$; 2nd dose, $15 \times 10^9$ CFU ml$^{-1}$) achieved a 100% complete response (CR) of tumours and substantially prolonged the survival rate of mice, likely owing to the double-dose regimen ensuring that a substantial amount of AUN was delivered to the tumours (Fig. 2d,e). No intolerable weight loss (>20%), designated as the ethical endpoint, was observed in any mouse after treatment with a single or double dose of AUN, indicating that the AUN toxicity was acceptable (Fig. 2f). In the colony assay, AUN showed tumour targeting, and the tumours completely disappeared from the mice within 240 h (Supplementary Fig. 6), achieving CR.

Other immunocompromised models included severe combined immunodeficient (SCID) and non-obese diabetic (NOD)-SCID mice, which were also used to further explore the potential anticancer efficacy of AUN without the assistance of the systemic immune system

(Fig. 2g). We expected that AUN would show strong anticancer efficacy in SCID and NOD-SCID models, similar to nude mice. SCID mice are characterized by an absence of functional T and B cells, lymphopenia, hypogammaglobulinaemia and a normal haematopoietic microenvironment[36]. However, NOD-SCID mice represent the same dysfunctions as SCID mice, with notably low activity of natural killer cells, macrophages and complement addition[37]. The single dose of AUN ($4.5 \times 10^9$ CFU ml$^{-1}$ for SCID and $3.0 \times 10^9$ CFU ml$^{-1}$ for NOD-SCID) achieved partial responses of Colon26 tumours in both SCID and NOD-SCID mice (Supplementary Fig. 7). Meanwhile, the double dose of AUN (1st dose, $1 \times 10^7$ CFU ml$^{-1}$; 2nd dose, $7 \times 10^9$ CFU ml$^{-1}$ for SCID, and 1st dose, $1 \times 10^8$ CFU ml$^{-1}$; 2nd dose, $15 \times 10^9$ CFU ml$^{-1}$ for NOD-SCID) surprisingly showed 100% CR of tumours and markedly increased the survival rate of mice for at least 30 days in both Colon26-bearing SCID and NOD-SCID mice (Fig. 2h–k). The control PBS (placebo) did not show any antitumour efficacy. We also confirmed that single and double doses of AUN did not cause intolerable weight loss in either immunocompromised model (Fig. 2l). These results indicated that AUN could serve as a useful therapeutic agent for effective tumour targeting and elimination in various immunocompromised models.

## Responses to the double dose that ameliorate lethality

Systemic immunological responses and intravascular cellular behaviours were subsequently investigated to clarify why the double-dose AUN administration was not lethal in mice and still had strong anticancer efficacy. The bacterial colonies of AUN were confirmed by the colony assay of blood samples extracted 5 min and 6 h after a single dose ($7.8 \times 10^9$ CFU ml$^{-1}$) or a double dose ($1 \times 10^7$ CFU ml$^{-1}$ and $7.8 \times 10^9$ CFU ml$^{-1}$) of AUN (Supplementary Fig. 8). The number of colonies obtained from the blood after a double dose of AUN was notably lower than that after a single dose of bacterial injection, especially at 6 h. These results suggest that AUN is effectively eliminated by immune cells. Flow cytometry analyses demonstrated that the number of neutrophils was substantially decreased in the spleen 2 days after the injection of a low dose of AUN ($1 \times 10^7$ CFU ml$^{-1}$) because neutrophils, which play a central role in early host defence following infection, were fought and consumed in the front line[38] against AUN invasion in the mouse body (Supplementary Fig. 9). In fact, the mice survived for at least 1 week even after administering a fatal dose of AUN ($15 \times 10^9$ CFU ml$^{-1}$) following neutrophil depletion using an anti-Ly6G antibody (Supplementary Table 2). Moreover, neutrophil depletion via anti-Ly6G antibody notably improved tumour clearance, even with a single low-dose AUN administration ($3 \times 10^9$ CFU ml$^{-1}$) and even with the administration of a combination of antiplatelet drugs (cilostazol,

acetylsalicylic acid and ticlopidine hydrochloride) and anticoagulation heparin (Supplementary Fig. 10). These results indicate that neutrophils may impede the spread of bacteria through the tumour and prevent complete oncolysis[39]. In any case, we consider that these systemic immunological eliminations of AUN could improve its safety and tumour-targeting effect, resulting in strong anticancer efficacy.

Various inflammatory cytokines, namely interferon-γ (IFNγ), tumour necrosis factor-α (TNF-α), interleukin-6, interleukin-1β and interleukin-17A, and the anti-inflammatory cytokine interleukin-10 were also expressed by both the single ($7.8 \times 10^9$ CFU ml$^{-1}$) and double doses of AUN (1st dose, $1 \times 10^7$ CFU ml$^{-1}$; 2nd dose, $7.8 \times 10^9$ CFU ml$^{-1}$ or 1st dose, $1 \times 10^7$ CFU ml$^{-1}$; 2nd dose, $15 \times 10^9$ CFU ml$^{-1}$) (Supplementary Figs. 11 and 12). Interestingly, inflammatory cytokines after the 2nd injection of the double dose of AUN ($7.8 \times 10^9$ CFU ml$^{-1}$) were somewhat suppressed compared with those after the single dose ($7.8 \times 10^9$ CFU ml$^{-1}$), probably because the low 1st dose of AUN ($1 \times 10^7$ CFU ml$^{-1}$) consumed aggressive immune cells such as neutrophils[38] to avoid releasing excess cytokines, as mentioned above.

Meanwhile, the levels of pathogen-sensitive immunoglobulins such as immunoglobulin G and immunoglobulin M in the blood samples did not substantially change 2 days after bacterial injection with a single low dose of AUN ($1 \times 10^7$ CFU ml$^{-1}$) (Supplementary Fig. 13). This result implies that adaptive immune responses to vaccination, which are generally triggered by B cells[40], had not yet been initiated in this short period (only 2 days).

By contrast, numbers of white blood cells and platelets (PLT) were notably changed by the single- or double-dose administrations of AUN, although other haematologic parameters—haematocrit, haemoglobin, mean corpuscular haemoglobin, mean corpuscular haemoglobin concentration, mean corpuscular volume and red blood cell count—were unchanged (Supplementary Figs. 14–17). Notably, after the first dose of the double-dose AUN administration, white blood cell count slightly increased compared with the standard value obtained at 0 h (approximately 4,700 µl$^{-1}$), while PLT count rapidly decreased over time. These results suggest that while the population of immature neutrophils increased, likely owing to the consumption of a large number of mature and trained neutrophils to fight AUN, leading to the replenishment of immature neutrophils, the blood coagulation factor PLT was also consumed to eliminate AUN after a low dose of AUN[41,42]. Therefore, we considered that the trade-off between immunological activation and thrombogenic PLT was an important factor for avoiding lethality in mice.

To verify which factor is the cause of death, whether it be direct bacterial virulence of AUN itself or AUN-mediated indirect thrombosis,

---

**Fig. 2 | Antitumour efficacy of AUN against various immunocompromised models. a**, Time course of establishment and treatment of BALB/c-nu/nu mice bearing Colon26 tumours. After tumour establishment, mice were intravenously injected with a single or a double dose of AUN. ca., circa. **b**, Images of mice after each treatment. NA, not available. **c**, In vivo anticancer effect of AUN. The suspension of AUN was intravenously injected once or twice into Colon26-bearing BALB/c-nu/nu mice. A control experiment (a single i.v. dose of PBS) was performed also. Data are represented as mean ± s.e.m.; $n = 5$ biologically independent mice. Statistical significance of the double-dose group at the endpoint was calculated by comparison with the PBS group. ****$P < 0.0001$ by Student's two-sided $t$-test. **d**, CR rate of Colon26 tumour-bearing BALB/c-nu/nu mice ($n = 5$ biologically independent mice) at 60 days after AUN (single- or double-dose) or PBS injection. ****$P < 0.0001$ by Student's two-sided $t$-test. **e**, Kaplan–Meier survival curves of Colon26 tumour-bearing BALB/c-nu/nu mice ($n = 5$ biologically independent mice) 150 days after tumour implantation. Statistical significance of the double-dose group at the endpoint was calculated by comparison with the PBS group. ****$P < 0.0001$ by log-rank (Mantel–Cox) test. **f**, Body weight measured daily after each treatment (single dose of AUN, double dose of AUN and single dose of PBS) in Colon26 tumour-bearing BALB/c-nu/nu mice. Data are represented as mean ± s.e.m.; $n = 5$ biologically independent mice. Statistical significance of the double-dose group at the endpoint was calculated

by comparison with the PBS group. ****$P < 0.0001$ by Student's two-sided $t$-test. **g**, Time course of establishment and treatment of Colon26 tumour-bearing SCID and NOD-SCID mice. After tumour establishment, mice were intravenously injected with a double dose of AUN. ca., circa. **h**, In vivo anticancer effect of the double dose of AUN against Colon26-bearing SCID (left) and NOD-SCID mice (right). The PBS or suspension of AUN was intravenously injected into the mice. Data are represented as mean ± s.e.m.; $n = 5$ biologically independent mice. Statistical significance at the endpoint was calculated by comparison with the PBS group. ****$P < 0.0001$ by Student's two-sided $t$-test. **i**, Images of mice after each treatment in Colon26-bearing SCID and NOD-SCID mice. **j**, CR rate of Colon26 tumour-bearing SCID and NOD-SCID mice ($n = 5$ biologically independent mice for each) 30 days after injection of bacteria or PBS. Statistical significance was calculated by comparison with the PBS group. ****$P < 0.0001$ by Student's two-sided $t$-test. **k**, Kaplan–Meier survival curves of Colon26 tumour-bearing SCID and NOD-SCID mice ($n = 5$ biologically independent mice) 30 days after tumour implantation. Statistical significance at the endpoint was calculated by comparison with the PBS group. ****$P < 0.0001$ by log-rank (Mantel–Cox) test. **l**, Body weight measured daily after various treatments in Colon26 tumour-bearing SCID and NOD-SCID mice. Data are represented as mean ± s.e.m.; $n = 5$ biologically independent mice. Statistical significance at the endpoint was calculated by comparison with the PBS group. ****$P < 0.0001$ by Student's two-sided $t$-test.

---

we investigated the survival of mice that received a fatal dose of AUN ($>7.8 \times 10^9$ CFU ml$^{-1}$) in the presence of various immunostimulants to actively eliminate AUN, these being polyinosinic:polycytidylic acid[43–45], granulocyte-macrophage colony-stimulating factor[46,47], vitamin D$_3$ (ref. 48) and imiquimod[49,50], and the anticoagulation drug heparin to prevent thrombosis[51] (Supplementary Table 2). Only heparin-administered mice survived after treatment with a single high dose of AUN

($15 \times 10^9$ CFU ml$^{-1}$); mice injected with immunostimulants after this dose died within 48 h, similar to the behaviour of mice without any immunostimulant administration. Herein we believed that the cause of death of a fatal dose of AUN in mice was not the pathogenicity of AUN but AUN-mediated thrombosis in blood vessels[38]. We also suspected that the cause of mouse mortality was severe cytokine release syndrome triggered by a high dose of AUN. Consequently, cytokine levels

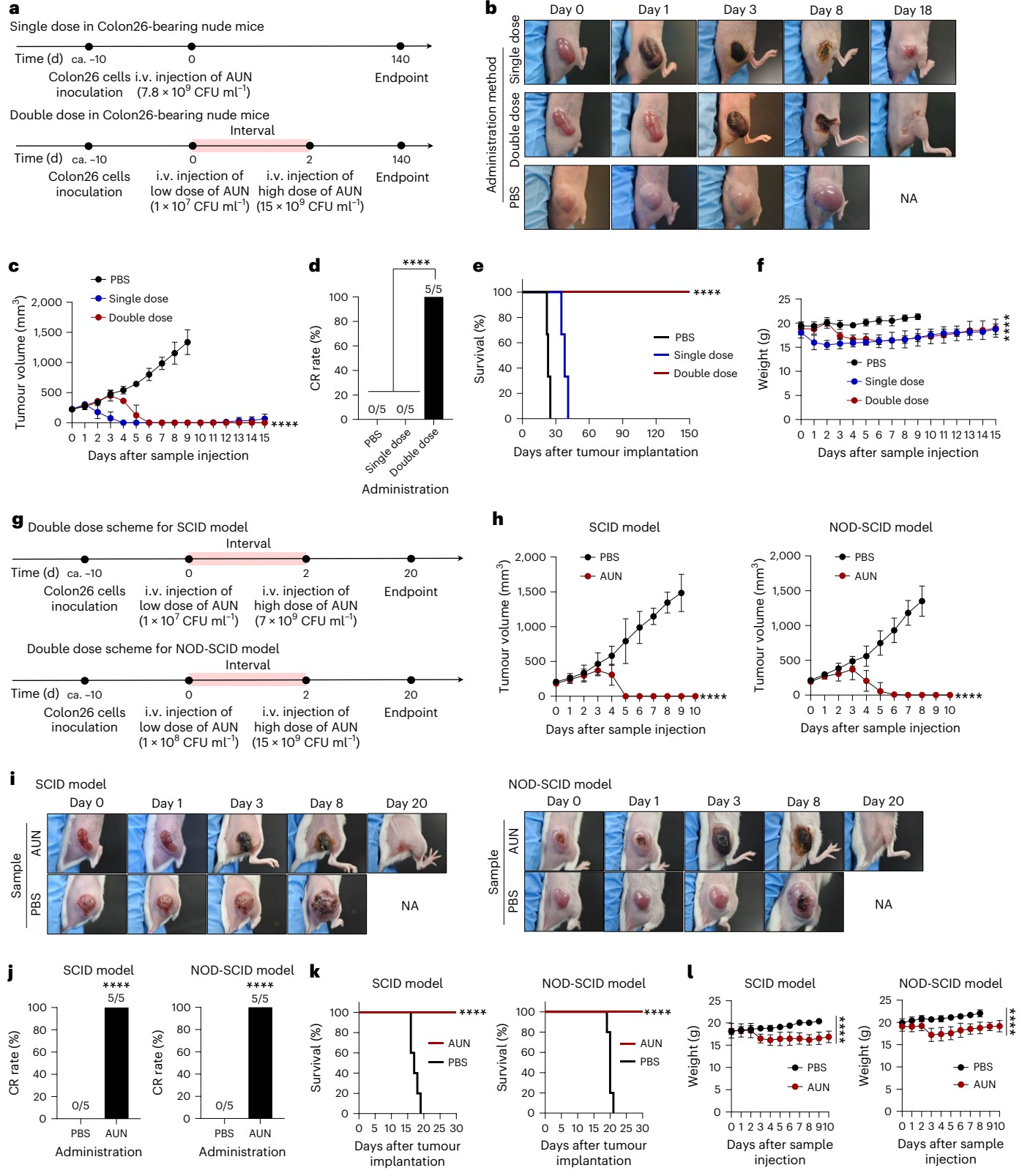

in heparin-administered mice were measured after treatment with a single high dose of AUN ($15 \times 10^9$ CFU ml$^{-1}$) (Supplementary Fig. 18). Predictably, cytokines in heparin-administered mice were suppressed after treatment with a single high dose of AUN ($15 \times 10^9$ CFU ml$^{-1}$) compared with the same dose of AUN-injected mice without heparin premedication (Supplementary Fig. 18). Indeed, heparin, used clinically as an anticoagulant, is known to have anti-inflammatory properties[52,53]. Although further investigations are required to determine whether thrombus causes death in mice, to clarify whether cytokine release syndrome is the main cause of death with a fatal dose of AUN in mice, the conventional anti-inflammatory agent (steroid) dexamethasone[54] was then tested (Supplementary Fig. 18 and Supplementary Table 2). Dexamethasone-premedicated mice survived after a single administration of a high dose of AUN ($15 \times 10^9$ CFU ml$^{-1}$). In addition, the cytokine levels in dexamethasone-treated mice were reduced after treatment with a single high dose of AUN ($15 \times 10^9$ CFU ml$^{-1}$). Notably, AUN showed stable and high antitumour efficacy under anti-inflammatory and immunosuppressive conditions induced by premedication with dexamethasone (Supplementary Fig. 19). Therefore, we concluded that death owing to a fatal dose of AUN in mice was mainly due to severe cytokine release syndrome.

Besides, clinical haematology and immunohistochemical (IHC) staining of vital organs confirmed that the double dose of AUN was safe in all of the mouse models that we evaluated—BALB/c-nu/nu, SCID and NOD-SCID—although biochemical parameters, such as alanine transaminase, aspartate aminotransferase, amylase and lactate dehydrogenase, in blood were somewhat influenced by the double dose of AUN, especially for the NOD-SCID mouse, likely owing to its vulnerable resistance to microbes derived from severe immunodeficiency (Supplementary Figs. 20–22 and Supplementary Tables 3–5). In addition, AUN did not show any toxicity in the tissues 150 days after i.v. injection (Supplementary Fig. 23). Haematoxylin and eosin (H&E) staining analyses demonstrated that the tissues after i.v. injection of AUN resembled those of the control group (PBS buffer). We further confirmed that AUN was safe in mice with lower limb ischaemia in an atherosclerosis-prone model (Supplementary Fig. 24 and Supplementary Table 6).

### Investigations of tumour regression mechanisms by AUN treatment

Next, we investigated the main cause of the strong antitumour efficacy of AUN in tumour-bearing immunocompromised mice using IHC staining, qPCR and flow cytometry. Surprisingly, all measurements taken by these three methods revealed that there were no active tumour-infiltrating immune cells in solid tumours of BALB/c-nu/nu immunocompromised mice for 24 h after treatment with AUN ($7.8 \times 10^9$ CFU ml$^{-1}$), although inflammatory cytokines such as TNF-α and IFNγ were observed in tumours, presumably owing to angiolysis and oncolysis triggered by bacterial invasion into the tumour[55–57] (Supplementary Figs. 25 and 26).

The tumours extracted 24 h after injection with AUN ($7.8 \times 10^9$ CFU ml$^{-1}$) were noticeably discoloured to dark red owing to AUN-mediated, highly tumour-selective thrombosis (Fig. 3a). The specific activity of factor VII, a blood-clotting protein involved in the coagulation cascade, was measured over time (Fig. 3b). Factor VII activity reached a plateau 24 h after AUN administration ($7.8 \times 10^9$ CFU ml$^{-1}$), when a dramatic colour change in tumours from the injected mice was observed (Fig. 2b).

To further investigate the mechanism by which the bacterial injection resulted in tumour suppression and tumour-specific thrombosis, H&E and IHC staining analyses were performed (Fig. 3c,d). Although further validation is required, we believe that the anticancer mechanism of double-dose AUN administration is similar to that of a single administration, as tumour-specific thrombosis was observed with both administrations. In the H&E staining assay, AUN-treated mice demonstrated obvious tumour damage with intercellular fragmentation,

indicating its antitumour effectiveness. Terminal deoxynucleotidyl transferase-mediated 2′-deoxyuridine, 5′-triphosphate nick end labelling (TUNEL) and TNF-α staining in tumour slices after treatment with AUN indicated massive apoptotic cell death and strong inflammatory responses. Fibrin, a blood coagulation protein, was also notably expressed in the slices from AUN-treated tumours. The control PBS (placebo) did not demonstrate any colour development in the staining for TUNEL, TNF-α or fibrin, although PBS-treated tumours also showed definitive pathologic features, including tight arrangement and nuclear atypia. Notably, AUN-treated mice showed a large number of thrombi mainly derived from PLTs in the blood vessels of the treated tumours with or without heparin premedication, likely because AUN affected the intratumoural vascular endothelial cells to recruit activated PLTs[58] (Fig. 3e). Besides, all tumour tissues were necrotic 24 h after i.v. injection of a single dose of AUN (200 μl, $5 \times 10^9$ CFU ml$^{-1}$) with and without heparin premedication (Supplementary Fig. 27). Tumour cells and blood vessels were intricately intertwined, and thrombi were widely and randomly observed 6 h after AUN administration (Supplementary Fig. 28). Thrombi are mainly composed of PLT aggregates, and neutrophils may be trapped. Blood vessels without thrombi are also observed. We believe that neutrophils are not a major factor in thrombus formation, as tumour-specific thrombi were observed after AUN administration following neutrophil depletion via anti-Ly6G, as mentioned above (Supplementary Fig. 10). Bacterial growth was observed just below the epidermal basal layer, in the tumour's internal vessel wall, in and around the blood vessel wall, and in the connective tissue at the base of the tumour, with or without heparin premedication (Supplementary Fig. 27). Moreover, our previous study demonstrated that intratumoural distributions of AUN cells and their colonies were evenly observed in all tumours by a microbial fluorescence in situ hybridization assay[29]. These results indicate that AUN effectively proliferates inside the tumour, simultaneously forming thrombi with the assistance of activated PLTs, and eventually necrotizes the whole tumour tissue.

All of the above cytokines, including TNF-α and IFNγ, could enlarge the intratumour vascular flow void, resulting in blood influx and tumour-specific thrombosis, with the help of PLT and fibrin[58,59]. We believe that the tumour-specific thrombosis caused by AUN could be used as an effective tumour antiangiogenesis strategy[38,60–62], as AUN-mediated tumour-specific thrombosis that tended to block the blood supply to tumours could cause hypoxia, which may fuel AUN proliferation and treatment effectiveness. Thrombosis induction in the tumour vasculature represents an appealing strategy to combat cancer. Coagulase from *Staphylococcus* somehow exhibits partial tumour growth inhibition[63]. Notably, anticancer efficacy of AUN is superior to that of *Staphylococcus* coagulase. Moreover, this report shows that such a natural microbial consortium, without any gene-engineering techniques, achieves a transcendent CR of tumours owing to the aid of native tumour-specific thrombosis. Interestingly, when heparin (5 mg per animal) was administered to Colon26-bearing BALB/c-nu/nu mice, the tumours presented an apparent change in colour and were effectively suppressed by a high dose of AUN ($15 \times 10^9$ CFU ml$^{-1}$) after 1 h, without any mice fatalities (Supplementary Fig. 29). This implies that thrombin and factor Xa, which are associated with the anticoagulation mechanism of heparin[52,53], do not influence tumour-specific thrombosis. Nevertheless, as mentioned earlier, we considered that tumour-specific thrombosis caused by PLT aggregations derived from the fibrinogen-activated PLT networks[64] 'ancillary' contributes to the strong antitumour efficacy of AUN in cooperation with the tumour-destructive intratumoural AUN proliferation, while maintaining the safety of AUN. As such, we further investigated the mechanism of thrombus formation and whether PLT plays an important role in tumour-targeted thrombosis. AUN induced tumour-specific discolouration and strong anticancer efficacy, even after dosing with various antiplatelet drugs or a combination of antiplatelet drugs and heparin (Supplementary Fig. 30). We confirmed that the tumour-specific

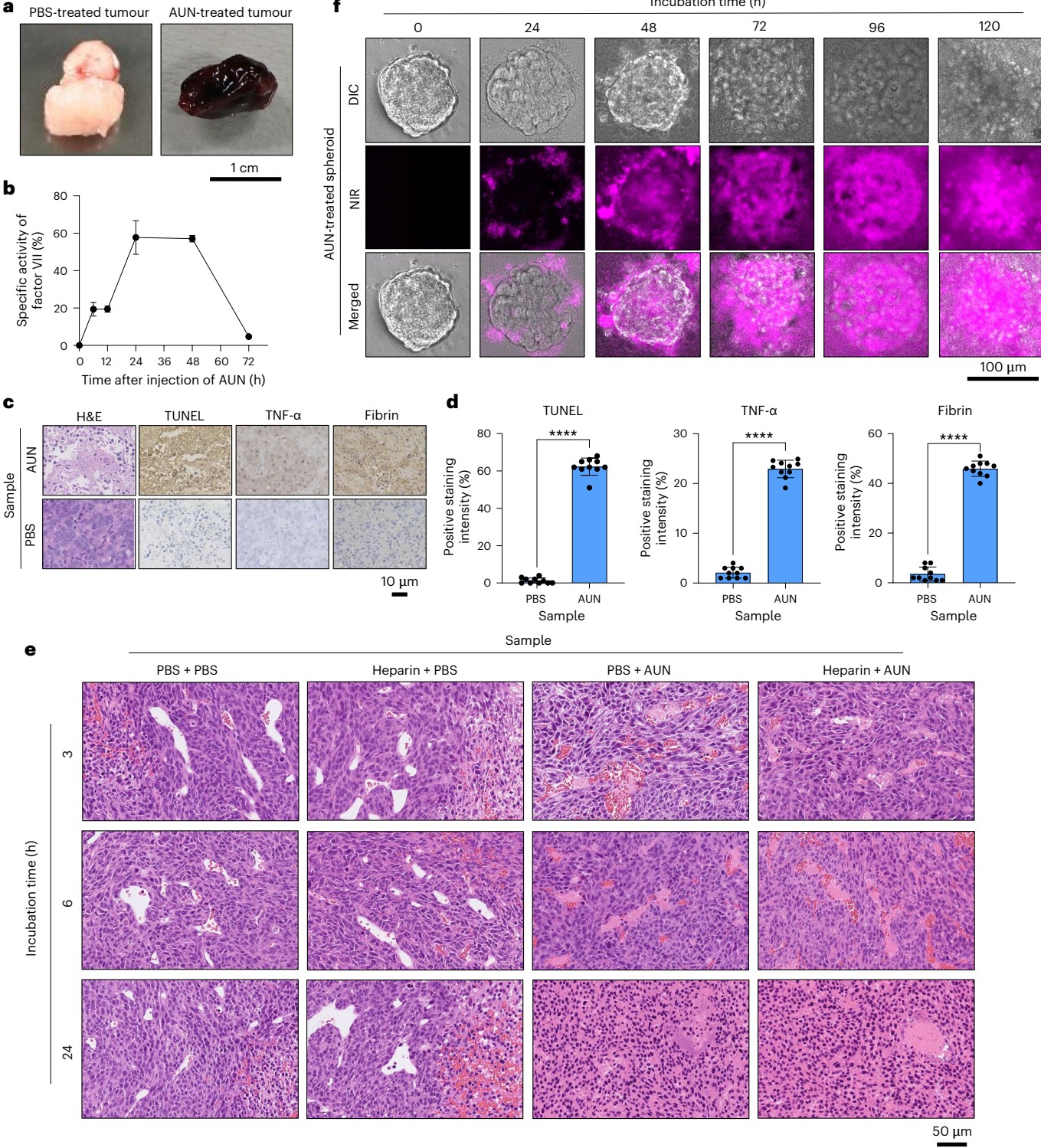

**Fig. 3 | Mechanism of tumour suppression in Colon26 tumour-bearing BALB/c-nu/nu mice by AUN. a**, Photos of tumours extracted 24 h after treatment with a single dose of PBS (left) or AUN ($7.8 \times 10^9$ CFU ml$^{-1}$) (right). **b**, Chronological specific activity of factor VII in blood serum after i.v. single injection of AUN ($7.8 \times 10^9$ CFU ml$^{-1}$). Specific activity of factor VII was calculated by comparison with the control group without any sample injections. Data are represented as mean ± s.e.m.; $n = 4$ independent experiments. **c**, Staining by H&E, TUNEL and IHC (for TNF-α and fibrin) in tumour tissues collected from different groups of mice at day 1 after a single dose of PBS or AUN ($7.8 \times 10^9$ CFU ml$^{-1}$). **d**, Statistical analyses of TUNEL-positive and IHC-stained (TNF-α and fibrin) tumour tissues in **c**. Data are represented as mean ± s.e.m.; $n = 10$ independent areas (regions of interest) in each tumour tissue collected from the groups of mice on day 1 after treatment with PBS or AUN. Statistical significance was calculated compared with the PBS group. ****$P < 0.0001$ by Student's two-sided $t$-test. **e**, Histology of tumours of PBS- or AUN-treated mice after administration with and without heparin. H&E-stained tumour tissues collected from different groups of mice at 3 h, 6 h and 24 h after respective treatments. PBS (200 µl) or AUN (200 µl, $5 \times 10^9$ CFU ml$^{-1}$) was intravenously injected to BALB/c-nu/nu mice ($n = 3$) after administration of heparin (5 mg per head) for 1 h. The negative control (PBS + PBS) was also performed without heparin premedication and AUN treatment. **f**, Fluorescence imaging of cancer spheroids incubated with AUN ($1 \times 10^7$ CFU) to evaluate its oncolytic ability ($n = 3$ independent experiments).

discolouration was caused by haematoma, not thrombi (Supplementary Fig. 31). These results indicate that PLT does not strongly affect antitumour efficacy or thrombus formation.

We further studied the oncolytic ability of AUN in a spheroid model originating from Colon26 cancer cells using optical microscopy with the near-infrared (NIR) fluorescent property of AUN[29,65,66] (Fig. 3f and Supplementary Fig. 32). This revealed that tumour spheroids were structurally destroyed over time after incubation with AUN, leading to the formation of a massive extracellular and intracellular biofilm of AUN both inside and outside the spheroids. We believe that tumour spheroid destruction was caused by AUN-secreted various cytolysins such as toxins (A-gyo_00232, A-gyo_00233, A-gyo_00346, A-gyo_01012, A-gyo_01238, A-gyo_01591, A-gyo_01802, A-gyo_02072, A-gyo_02290, A-gyo_02470, A-gyo_02471, A-gyo_02779, A-gyo_02886, A-gyo_02897, A-gyo_03121, A-gyo_03468, A-gyo_03469, UN-gyo_00877, UN-gyo_01182, UN-gyo_01266 and UN-gyo_03065), haemolysins (A-gyo_00205, A-gyo_00206, A-gyo_00231, A-gyo_00311, A-gyo_02020, UN-gyo_01267 and UN-gyo_04003), phospholipases (A-gyo_00091, A-gyo_02735, A-gyo_02737 and UN-gyo_04450) and adenylate cyclases (A-gyo_02748 and A-gyo_01357) (Supplementary Data 1 and Supplementary Data 2). To investigate whether these bacteria-secreted toxins and enzymes have anticancer efficacy, we performed cytotoxicity tests and observed spheroid destruction behaviour using homogenized AUN (dead cells) (Supplementary Fig. 33). As expected, homogenized AUN showed strong anticancer and oncolytic activities owing to its cytotoxic toxins and enzymes.

In vitro anticancer efficacy of AUN was also evaluated. After co-culturing with Colon26 cells for 24 h, AUN showed strong cytotoxicity at several bacterial concentrations because of the various cytolytic exotoxins from AUN. Conversely, normal human diploid fibroblast (MRC5) cells were somewhat tolerant to AUN. Radioimmunoprecipitation assay lysis buffer was used as a positive control and showed strong cytotoxicity in both cell lines (Fig. 4a). AUN showed the highest cytotoxicity against Colon26 compared with A-gyo and UN-gyo (Supplementary Fig. 34). Optical microscopy of the bacteria at any AUN concentration after this 24 h incubation revealed a wondrous fibrous transformation of A-gyo from a short swimmer cell (approximately 2.5 µm) to a hyper-elongated swarmer cell (approximately 20–50 µm) (Fig. 4b,c). Using its flagellae, ca-PM showed the same unique transformation ability on a solid surface, and its motile swarmer cell differentiation paralleled the increased expression of several virulence factors[24]. We confirmed that UN-gyo does not directly contribute to the transformation of the fibrous structure of A-gyo by fluorescence microscopy (Supplementary Fig. 35). The fluorescent pink dots derived from UN-gyo were not embedded in the A-gyo fibres. Notably, the A-gyo transformation was specifically observed only when AUN was co-cultured with Colon26 cells, instead of MRC5, likely owing to the bacteria's selective sensitivity to a chemical signal emitted by cancer cells[67] (Supplementary Fig. 36a). Meanwhile, A-gyo itself non-specifically formed fibrous structures on both Colon26 and MRC5 cells, although there were more A-gyo swarmers in Colon26 than in MRC5. AUN showed a larger number of fibrous swarmers after co-culturing with Colon26 cells rather than after treatment with A-gyo. In contrast to AUN and A-gyo, UN-gyo did not show distinct massive and long fibres after co-culturing with either Colon26 or MRC5 cells. Interestingly, various oncometabolites[68], such as fumaric acid, lactic acid, O-phosphorylethanolamine and spermidine, facilitated A-gyo transformation in the culture medium when AUN was co-cultured with each oncometabolite molecule (Supplementary Fig. 36b). These results indicate that AUN bacterial consortium has a more selective fibrous transformation ability for A-gyo swimmers against cancer cells than pure bacterial strains A-gyo and UN-gyo. Swarmer cell differentiation (a large number of fibrous constructs derived from A-gyo) and massive thrombosis were also confirmed in the blood vessels of the tumour tissues of Colon26-bearing nude mice 24 h after the i.v. injection of AUN (Fig. 4d). In addition, the active movements of numerous short swimmers and transformed swarmer cells of A-gyo were observed throughout the AUN-treated tumour tissues, presumably because such highly motile A-gyo could deeply penetrate the inner core of the tumour tissues (Fig. 4e,f and Supplementary Video 1). NIR fluorescence derived from UN-gyo was confirmed only in limited regions, including the edges of tumour tissues and the tumour tissues close to the damaged and shrunken blood vessels, probably owing to the low UN-gyo motility (Fig. 4g). By contrast, the control PBS (placebo) group did not exhibit any fibres, motile bacteria or severe thrombosis (Supplementary Fig. 36c and Supplementary Video 2). However, the coagulated blood samples showed no fibrous constructs (Supplementary Fig. 36c). Therefore, we consider that the active swimmer and fibrous and motile swarmer forms of A-gyo mainly express not only the effective elimination of cancer cells in the immunocompromised tumour microenvironment but also their highly tumour-specific angiolysis in blood vessels. Notably, haemolysin-associated genes were expressed in all tested bacterial strains (Fig. 4h). While A-gyo and UN-gyo exhibited β-haemolysis, AUN showed no haemolysis on blood agar plates (Fig. 4i). We also confirmed that haemolysis was not observed in human blood cells after treatment with AUN at any concentration tested (Supplementary Fig. 37). This result indicates that haemolytic A-gyo and UN-gyo may attenuate toxicity when they coexist with AUN. In addition, all bacterial strains, in addition to the homogenized AUN (dead cells), did not have any coagulation properties because they did not secrete procoagulant molecules (for example, coagulase and thrombin-like enzymes[69,70]) (Supplementary Fig. 38). Furthermore, qPCR analysis said that notable changes in the bacterial ratio of AUN (A-gyo:UN-gyo ≈ 99:1) were observed in a tumour 24 h after i.v. AUN administration (the original ratio is 'naturally' adjusted to A-gyo:UN-gyo ≈ 3:97) (Supplementary Fig. 39). Therefore, we believe that outstanding anticancer performance and high safety level of AUN are driven by the dramatic bacterial population shift-triggered reactivation of the suppressed angiolytic and oncolytic activities of AUN in tumours. More interestingly, thrombus formation and destruction of blood vessels in the targeted tumours could be directly observed over time after the i.v. AUN injection, likely owing to the inherent bacterial haemolysis of the reactivated A-gyo by a population shift against intratumoural red blood cells and fragile tumour blood vessels[71] (Fig. 4j, Supplementary Fig. 40 and Supplementary Videos 3–5). NIR-II fluorescence imaging[72] of IR-1061-modified AUN also demonstrated that AUN was distributed in the tumour over time in accordance with tumour discolouration owing to bacterial destruction of intratumour vascular networks (Supplementary Fig. 41). Meanwhile, as we mentioned earlier, iron is essential for various cellular functions in cancer, and iron depletion strategies have a strong anti-proliferative effect on tumour cells[32–34]. We found that transferrin and ferritin in the tumour microenvironment were certainly depleted by AUN treatment, presumably owing to the increase in iron requirements of AUN via bacterial cross-talk (as indicated by the transcriptome analysis) (Supplementary Fig. 42).

In summary, we conclude that the transcendent anticancer efficacy of AUN is mainly caused by tumour-specific vascular destruction owing to unique bacterial features, such as transformation and haemolytic actions, through an intratumoural bacterial population shift (Fig. 4k). The selective destruction of tumour cells by cytolytic bacterial exotoxins can strongly influence tumour suppression. The exhaustion of iron in the tumour milieu, owing to the increased iron requirements of AUN, is also involved in tumour suppression.

## Antitumour efficacy of AUN in immunocompromised models bearing human cancer cells

Finally, we examined the therapeutic efficacy of AUN in various immunocompromised models derived from human cancer cell lines. The therapeutic AUN performance at a combination of low and high doses was assessed in BALB/c-nu/nu mice with human colorectal

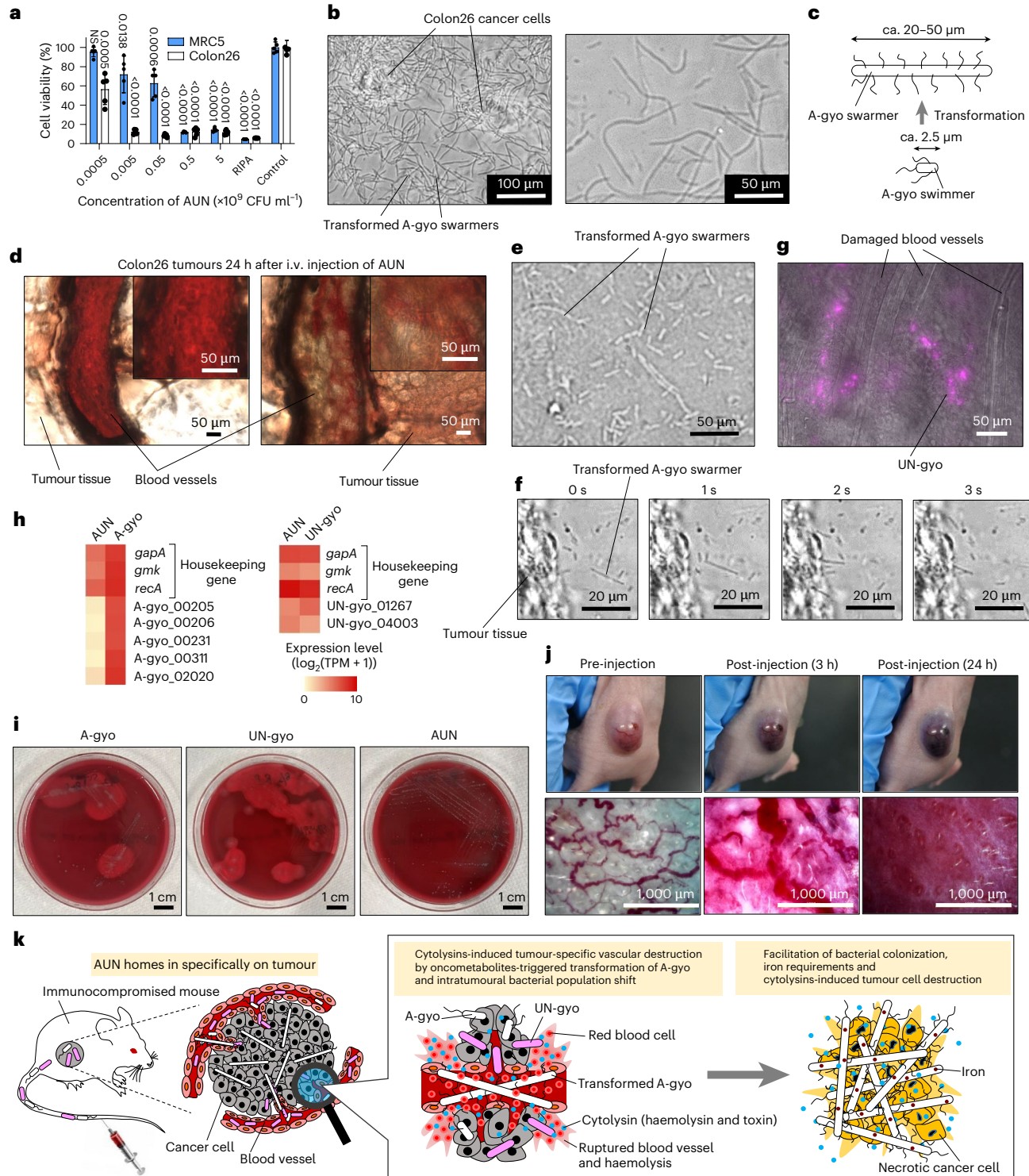

**Fig. 4 | Effects of bacterial features of AUN on antitumour efficacy.**
**a**, Cytotoxicity of AUN. The viabilities of Colon26 and MRC5 cells were tested
24 h after treatment with AUN at different bacterial concentrations. Data
are represented as mean ± s.e.m.; $n = 5$ independent experiments. Statistical
significance was calculated compared with the control group without any
sample treatment. NS ($P = 0.2044$); $P$ values were obtained using Student's
two-sided $t$-test. **b**, Optical microscopic images (left, low magnification; right,
high magnification) of transformed A-gyo swarmers after co-culturing of AUN
($5 \times 10^6$ CFU ml$^{-1}$) with Colon26 cells for 24 h at 37 °C in a humidified incubator
containing 5% $CO_2$ ($n = 3$ independent experiments). **c**, Schematic illustration
of transformation of A-gyo from a swimmer cell to a swarmer cell. ca., circa.
**d**, Optical microscopic images of blood vessels in the tumour tissues 24 h after

i.v. injection of AUN ($7.8 \times 10^9$ CFU ml$^{-1}$, 200 µl) into Colon26 tumour-bearing
BALB/c nude mice ($n = 3$ independent experiments). **e**, Optical microscopic
image of A-gyo swarmers in the tumour tissue ($n = 3$ independent experiments).
**f**, Real-time observation of the movement of a motile A-gyo swarmer in the
tumour tissue ($n = 3$ independent experiments). **g**, Fluorescence image of UN-
gyo in the tumour tissue close to the damaged blood vessels ($n = 3$ independent
experiments). **h**, Heat map for the haemolysin-associated gene expression level
of AUN/A-gyo (left) and AUN/UN-gyo (right) based on the $\log_2$ of TPM. **i**, Photos
of haemolysis behaviours of A-gyo, UN-gyo and AUN on blood agars. **j**, Real-
time observation of the destruction of intratumoural blood vessels in Colon26
tumour-bearing BALB/c nude mice after i.v. injection of AUN ($7.8 \times 10^9$ CFU ml$^{-1}$,
200 µl). **k**, Scheme of the proposed mechanism.

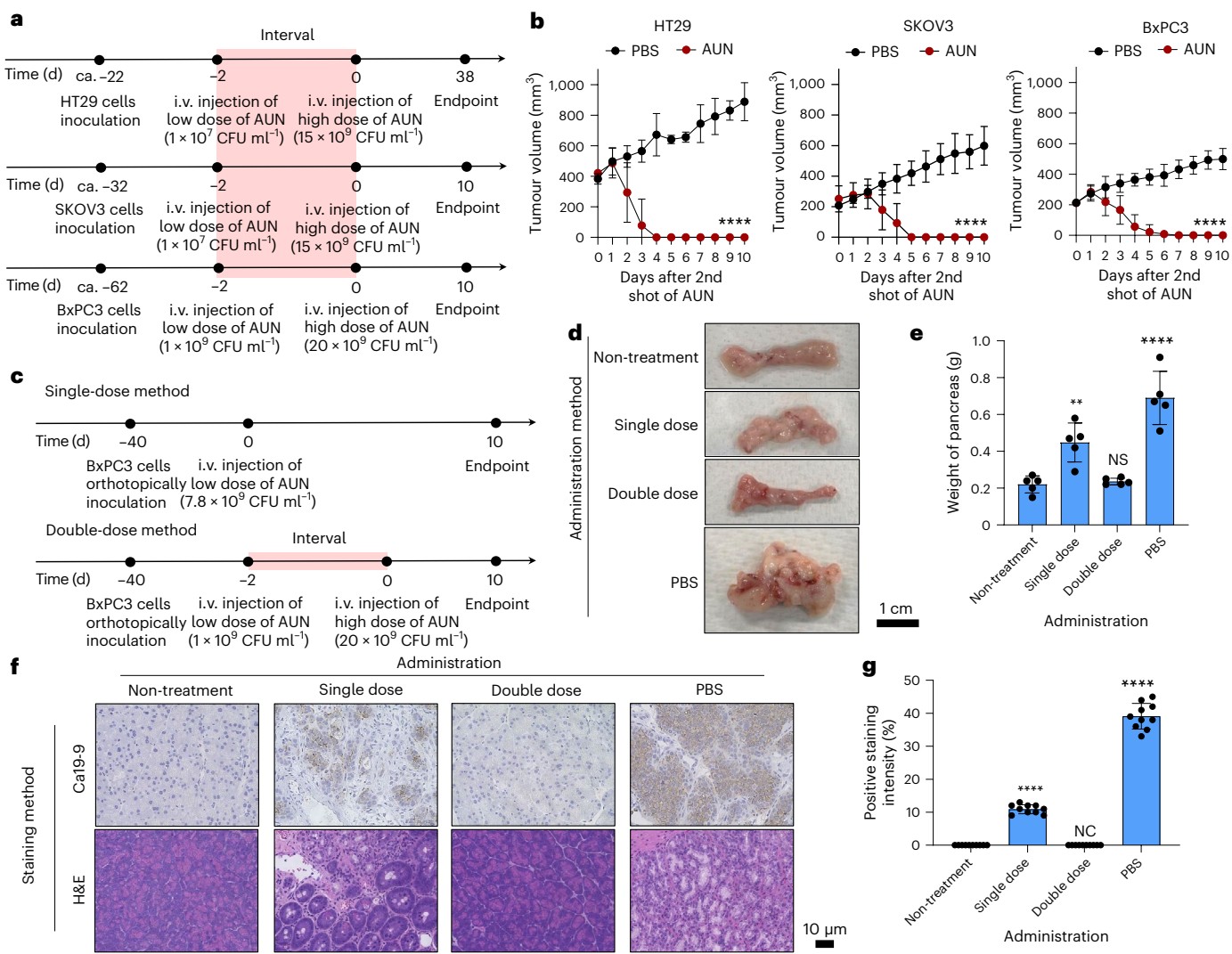

**Fig. 5 | Antitumour efficacy of AUN against various types of mouse bearing human tumours. a**, Time course of establishment and treatment of BALB/c-nu/nu mice bearing various human tumours (HT29, SKOV3 and BxPC3). After tumour establishment, mice were intravenously injected with PBS or a double dose of AUN. ca., circa. **b**, In vivo anticancer effect of the double dose of AUN against HT29-, SKOV3- and BxPC3-bearing BALB/c-nu/nu mice. PBS or a suspension of AUN was intravenously injected into the mice. Data are represented as mean ± s.e.m.; $n$ = 5 biologically independent mice. Statistical significance was calculated by comparison with the PBS group. ****$P$ < 0.0001 by two-way analysis of variance. **c**, Time course of establishment and treatment of BxPC3 tumour-bearing orthotopic BALB/c-nu/nu mice. Mice were intravenously injected with PBS or with a single or double dose of AUN. **d**,**e**, Visual appearance (**d**) and weight of

pancreases (**e**) extracted 10 days after each treatment. Data are represented as mean ± s.e.m.; $n$ = 5 independent experiments. Statistical significance was calculated by comparison with an untreated group. NS ($P$ = 0.4929), **$P$ = 0.022 and ****$P$ < 0.0001 by Student's two-sided $t$-test. **f**, IHC (for Ca19-9)- and H&E-stained tumour tissues collected from different groups of mice at day 10 after respective treatments. **g**, Statistical analyses of IHC (Ca19-9) in **f**. Data are represented as mean ± s.e.m.; $n$ = 10 independent areas (regions of interest) in each tumour tissue collected from the groups of mice on day 10 after treatment with PBS or with a single or double dose of AUN, or after sham treatment. Statistical significance was calculated by comparison with the non-treatment group. NC, not calculable (statistical testing could not be performed because all values in both groups were identical), and ****$P$ < 0.0001 by Student's two-sided $t$-test.

---

adenocarcinoma (HT29), human ovarian cancer (SKOV3) and human pancreatic cancer (BxPC3) (Fig. 5a). All tested tumours successfully disappeared without intolerable side effects in all tested tumour models, although slight differences in anticancer reactivity among tumour types were observed (Fig. 5b and Supplementary Figs. 43–45). These differences in the speed of antitumour responses against each cancer type, including the haemorrhagic Colon26, may be related to the intratumoural expression of inflammatory cytokines (Supplementary Fig. 46). For instance, inflammatory cytokines in the highly responsive Colon26 cells were strongly expressed, rather than in the gradually responsive BxPC3. The AUN bacterial consortium exhibits transcendent oncolytic ability against various mouse- and human-derived malignant tumours in immunocompromised models.

In addition, the orthotopic graft model BxPC3 was prepared for treatment with a single ($7.8 \times 10^9$ CFU ml$^{-1}$) or double dose of AUN (1st dose, $1 \times 10^9$ CFU ml$^{-1}$; 2nd dose, $20 \times 10^9$ CFU ml$^{-1}$) (Fig. 5c). PBS was used as the placebo control for AUN. The untreated group was also tested as a negative control without BxPC3 tumour inoculation or any sample administration. In particular, a double dose of AUN was dramatically effective against the orthotopic BxPC3-bearing tumour model (Fig. 5d,e). IHC staining of Ca19-9 as a biomarker, predictor and promoter of pancreatic cancer also showed that the tumour tissues of mice administered a double dose of AUN appeared similar to those of the non-treatment group, whereas colour changes were observed in the groups administered a single dose of AUN or PBS (Fig. 5f,g).

## Discussion

We demonstrated an approach using the intratumoural bacterial consortium AUN, which can preferentially grow and proliferate within a targeted tumour milieu, for the treatment of various immunocompromised mouse and human cancer models. Given the therapeutic characteristics of AUN, outstanding efficacy and excellent life prolongation have been achieved in mouse models of subcutaneous and orthotopically transplanted tumours. In addition, we discovered that a fatal dose of AUN could be ameliorated by a simple administration method using a combination of low and high doses to achieve stronger efficacy by mitigating cytokine release syndrome in blood vessels. Moreover, by unveiling mechanistic insights, we showed that the AUN anticancer efficacy was driven by its unique bacterial features and tumour-specific oncolysis, without the assistance of cytotoxic immune cells in the tumour microenvironment. AUN performance is potentially superior to conventional treatments, including bacteria-based therapies and immuno-oncology drugs, especially in terms of anticancer efficacy, tumour targeting, need for biomarkers, production cost, safety and necessity of genetic engineering. Notably, high AUN efficacy can be achieved by simple double-dose administration. Ordinary anticancer immunotherapies, including previously engineered bacteria-mediated remedies, immune checkpoint blockades, monoclonal antibodies and CAR-T cell therapy, do not achieve transcendent efficacy in immunocompetent or immunocompromised models. Therefore, the AUN-based bacterial approach may revolutionize cancer therapy and advance the concept of using microbial modalities for therapeutics based on a unique mechanism of action regardless of the presence or absence of healthy immunological systems. These findings also highlight the potential for further studies to enhance the efficacy of AUN and develop a premedication regimen for future clinical trials. In addition, although transient immune clearance was observed following AUN administration, as evidenced by the results of various safety tests, the health conditions of the treated mice returned to normal because of strong homeostatic immune resilience. However, we envision that a long-term and careful assessment of immunobiological and physiological factors in large animals is essential for future clinical trials. However, genetic engineering[10–15] and our recently reported scaffold-mediated bacterial culturing method[73] are useful for toxicity attenuation and could be applied to the development of safe and secure bacterial modalities. Thus, the potential of AUN as a blockbuster drug for cancer treatment is promising.

## Methods

### Animal experiments

The animal experiments were conducted following the protocols approved by the Institutional Animal Care and Use Committee of Japan Advanced Institute of Science and Technology (number 07-001). For the in vivo studies, the animals were randomly assigned to treatment groups at the outset of the study. Mice were assigned consecutive numbers across cages, and the groups were clustered consecutive numbers at random. All animals were sex- and age-matched within each experimental group. Different experiments used animals of different ages as appropriate. Male mice were used only in the BxPC3 orthotopic pancreatic tumour model; all other experiments used female mice. No animals were excluded based on sex or age, and no selection bias was introduced in assigning animals to treatment groups. Mice were group-housed in ventilated clear plastic cages under appropriate ambient temperature (~23 °C), humidity (~50%) and standard 12 h/12 h light/dark conditions. Experimental group sizes were approved by the regulatory authorities for animal welfare after being defined to balance statistical power, feasibility and ethical aspects. The maximal tumour burden permitted is 3,000 mm³, and the maximal tumour size/burden was not exceeded in the experiments. No animals or data points were excluded from the analyses.

### Bacterial strains and growth

The bacterial strains A-gyo and AUN tested in this study were isolated from mouse colon carcinoma (Colon26)-derived tumours. BALB/cCrSlc mice (female; 4 weeks old; average weight 15 g; $n$ = 8; genetic background, BALB/c; substrain, BALB/cCrSlc-nu/nu) were obtained from Japan SLC. Mice bearing the Colon-26 cell-derived tumours were generated by injecting 100 µl of the culture medium/Matrigel (Dow Corning) mixture (v/v = 1:1) containing $1 \times 10^6$ cells into the right side of the backs of the mice. After approximately 20 days, when the tumour volumes reached ~400 mm³, the mice were intravenously injected with 200 µl culture medium containing commercially available *R. palustris* (ca-RP) (catalogue number NBRC 16661, National Institute of Technology and Evaluation) ($5 \times 10^9$ CFU ml⁻¹). After 2 days, the tumours were carefully excised. After homogenizing thoroughly with a homogenizer pestle in 1 ml of PBS (catalogue number 164-23551, FUJIFILM Wako Pure Chemical) at 4 °C, the mixture was shaken for 20 min at a speed of $10 \times g$ at 15 °C. The supernatant was diluted 10 times with PBS, and a sample (100 µl) was inoculated onto an agar plate. After incubation for 3 days under anaerobic conditions and tungsten lamps, the red bacterial colonies derived from *R. palustris* were formed on plates. Several colonies were picked and cultured anaerobically in ATCC 543 medium at 26–30 °C under tungsten lamps for 3 days. Grey-coloured bacterial solution was subcultured in conventional ATCC 543 medium or ATCC 543 medium without L-cysteine (Cys) (catalogue number 039-20652, FUJIFILM Wako Pure Chemical). Subcultured red bacterial suspension prepared in ATCC 543 medium without Cys was disseminated onto an ATCC 543 agar plate with 0.1% sodium deoxycholate (catalogue number 190-08313, FUJIFILM Wako Pure Chemical) and without Cys. After 7 days, red and white colonies were observed on the plate. A single red colony was carefully picked up using an inoculating loop and then cultured in an ATCC 543 medium without Cys for 3 days. A prepared red bacterial solution (AUN) was stocked and used for further tests.

After subculturing, the obtained grey-coloured bacterial solution prepared using conventional ATCC 543 medium was inoculated onto an ordinary ATCC 543 agar plate containing 0.1% sodium deoxycholate. White bacterial colonies were formed on the plate. A single white colony was carefully picked up by an inoculating loop and then anaerobically cultured in conventional ATCC 543 medium at 26–30 °C under tungsten lamps for 3 days. A prepared grey bacterial solution (A-gyo) was stocked and used for further tests.

Besides, a single red colony prepared from AUN solution was carefully picked up using a 29-gauge syringe needle (0.337 mm diameter, Myjector Insulin Syringe; Terumo) under stereo microscopy (M2-1139-01; AS ONE), and then anaerobically cultured in an ATCC 543 medium without Cys at 26–30 °C under tungsten lamps for 3 days. A prepared red solution (UN-gyo) was stocked and used for further tests.

A-gyo cells were anaerobically cultured in ATCC 543 medium containing 0.06% L-cysteine at 26–30 °C under tungsten lamps. UN-gyo, AUN and ca-RP cells were also grown anaerobically in liquid cultures at 26–30 °C in ATCC 543 medium under tungsten lamps. Commercially available *P. mirabilis* (ca-PM) (catalogue number ATCC 35659, American Type Culture Collection) was anaerobically cultured at 30 °C in NBRC 802 medium using an incubator (i-CUBE FCI-280HG; AS ONE). Commercially available *Staphylococcus aureus* (catalogue number NBRC 102135, National Institute of Technology and Evaluation) was aerobically cultured at 35 °C in NBRC 802 medium using an incubator. ATCC 543 medium and NBRC 802 medium were prepared according to the cell bank's instructions. The number of bacterial cells and their viability were measured using a bacterial counter (CASY Cell Counter & Analyser; OMNI Life Science) in addition to an active colony assay.

To observe haemolysis behaviours, suspensions of AUN, A-gyo or UN-gyo were streaked on blood agar plates (TSA II 5% Sheep Blood Agar M; catalogue number 252202, Becton, Dickinson and Company) using an inoculation loop, and anaerobically cultured for 3 days.

## Characterization of bacteria

The morphology and structure of A-gyo and ca-PM were observed under a high-resolution transmission electron microscope (JEM-1400Flash; JEOL) at a 100 kV acceleration voltage after negative staining by 2% uranium acetate (catalogue number 02624-AB, SPI Supplies). The microscopic observations were performed by the Hanaichi UltraStructure Research Institute. For microscopic imaging of bacteria, bacterial solutions (20 μl, $5 \times 10^8$ CFU ml⁻¹) of A-gyo or ca-PM were plated on glass coverslips (AGC Techno Glass) and then observed using a fluorescence microscopy system (IX73; Olympus) and cellSens V3.1 software (Olympus) equipped with objectives (×60 magnification, aperture 1.35; UPLSAPO60X, Olympus or ×100 magnification, aperture 0.95; PLFLN100X, Olympus) at 20 °C.

For the bacterial swarming motility test, A-gyo or ca-PM suspension (10 μl, $4 \times 10^6$ CFU) was inoculated onto an ATCC 543 agar plate. After incubation for 5 days under anaerobic conditions at 26 °C using an incubator, the bacterial colonies derived from A-gyo or ca-PM were formed on the plates. Kinematic analyses of A-gyo and ca-PM were performed using a light microscopy system (BZ-X810; Keyence) and dynamics analysis applications software (BZ-X800 Analyser V1.1.2.4; Keyence).

Comparative genome analyses and transcriptome analyses of bacteria were performed by Rhelixa and Bioengineering Lab, respectively.

Minimal inhibitory concentrations of antibiotics against A-gyo, UN-gyo and AUN were determined at 600 nm (OD600) using a microplate reader (Infinite M200 PRO; Tecan) at different concentrations of antibiotics (0 μg ml⁻¹, 0.05 μg ml⁻¹, 0.1 μg ml⁻¹, 0.2 μg ml⁻¹, 0.4 μg ml⁻¹, 0.8 μg ml⁻¹, 1.6 μg ml⁻¹, 3.1 μg ml⁻¹, 6.3 μg ml⁻¹, 12.5 μg ml⁻¹, 25 μg ml⁻¹ and 50 μg ml⁻¹). Ampicillin (catalogue number 016-23301), cefotaxime (catalogue number 034-16111), imipenem (catalogue number 092-07281), minocycline (catalogue number 135-18671) and vancomycin (catalogue number 226-01301) were purchased from FUJIFILM Wako Pure Chemical. Aztreonam (catalogue number A2466), amoxicillin (catalogue number A2099), gentamicin (catalogue number G0383), erythrosine (catalogue number E0751), clindamycin (catalogue number C2257) and levofloxacin (catalogue number L0193) were obtained from Tokyo Chemical Industry.

## Cell culture

The Colon26 cell line (catalogue number RBC2657) was obtained from RIKEN BioResource Research Center. Human normal diploid fibroblast (MRC5) cell line (catalogue number JCRB9008) was obtained from the Japanese Collection of Research Bioresources Cell Bank. Human pancreatic cancer (BxPC3) cell (catalogue number CRL1687) was obtained from American Type Culture Collection. Human ovarian cancer (SKOV3) (catalogue number EC91091004-F0) and human colorectal adenocarcinoma (HT29) (catalogue number EC91072201-G0) cells were obtained from KAC. All cell lines except for MRC5 were cultured in Roswell Park Memorial Institute 1640 Medium (catalogue number 30264-56, Nacalai Tesque) containing 10% fetal bovine serum (catalogue number S1810-500, Biowest) and penicillin–streptomycin–amphotericin B (catalogue number 161-23181, FUJIFILM Wako Pure Chemical) (concentrations of penicillin, streptomycin and amphotericin B are 100 IU ml⁻¹, 0.1 mg ml⁻¹ and 0.25 μg ml⁻¹, respectively). MRC5 cell was cultured in Dulbecco's modified Eagle's medium (catalogue number 08458-16, Nacalai Tesque) containing 10% fetal bovine serum and penicillin–streptomycin–amphotericin B (concentrations of penicillin, streptomycin and amphotericin B are 100 IU ml⁻¹, 0.1 mg ml⁻¹ and 0.25 μg ml⁻¹, respectively). Cells were maintained at 37 °C in a humidified incubator containing 5% $CO_2$, then cryopreserved in multiple vials and stored in liquid nitrogen. Cell stocks were regularly revived to avoid the genetic instabilities associated with high passage numbers.

## Antitumour therapy

The Colon26 tumour-bearing BALB/c-nu/nu mice (female; about 7 weeks old; average weight = 20 g; average tumour size ~200 mm³; n = 5; genetic background, BALB/c; substrain, BALB/cCrSlc-nu/nu; obtained from Japan SLC) were intravenously injected in the tail vein with culture medium (200 μl) containing AUN ($1 \times 10^7$ CFU ml⁻¹ or $7.8 \times 10^9$ CFU ml⁻¹). Culture medium (200 μl) containing AUN ($15 \times 10^9$ CFU ml⁻¹) was intravenously injected again 2 days after administration of AUN ($1 \times 10^7$ CFU ml⁻¹). Control experiments were also performed using PBS (200 μl). The tumour formation and overall health (viability and body weight) were monitored daily. The tumour volume was calculated using $V = L \times W^2/2$, where $L$ and $W$ denote the length and width of the tumour, respectively. The survival ratio of Colon26 tumour-bearing mice (n = 5 biologically independent mice) was also measured for 60 days after treatment. When the tumour volumes reached more than 1,500 mm³, the mice were euthanized at the endpoint according to the approved protocols.

To investigate various immunostimulants, such as polyinosinic:polycytidylic acid, granulocyte-macrophage colony-stimulating factor, vitamin $D_3$ and imiquimod, for boosting the immune system, BALB/c-nu/nu mice (female; about 6 weeks old; average weight = 18 g; n = 5; genetic background, BALB/c; substrain, BALB/cCrSlc-nu/nu; obtained from Japan SLC) were intravenously injected in the tail vein with D-PBS (catalogue number 14249-24, Nacalai Tesque) (200 μl) containing polyinosinic:polycytidylic acid (catalogue number 550-28211, FUJIFILM Wako Pure Chemical) (200 μg per head) or recombinant mouse granulocyte-macrophage colony-stimulating factor from *Escherichia coli* (catalogue number 17366-07, Nacalai Tesque) (1 μg per head), or 1% Cremophor EL (catalogue number 09727-14, Nacalai Tesque) in D-PBS (200 μl) containing vitamin $D_3$ (catalogue number 36403-91, Nacalai Tesque) (200 μg per head). Only imiquimod (hydrochloride) (catalogue number HY-B0180A, FUJIFILM Wako Pure Chemical) (400 μg per head) was intraperitoneally injected to mice. Culture medium (200 μl) containing AUN ($15 \times 10^9$ CFU ml⁻¹) was then intravenously injected 2 days after administration of each immunostimulant.

The anticoagulant heparin was used to identify whether thrombosis caused by AUN could affect the viability of mice. AUN (200 μl, $15 \times 10^9$ CFU ml⁻¹) was intravenously injected into BALB/c-nu/nu mice (female; about 6 weeks old; average weight = 18 g; n = 5; genetic background, BALB/c; substrain, BALB/cCrSlc-nu/nu; obtained from Japan SLC) 1 h after administration of D-PBS (200 μl) containing heparin sodium salt (catalogue number 17513-41, Nacalai Tesque) (5 mg per head). To study the effect of heparin on tumour efficacy, Colon26 tumour-bearing BALB/c-nu/nu mice (female; about 8 weeks old; average weight = 20 g; average tumour size ~200 mm³; n = 5; genetic background, BALB/c; substrain, BALB/cCrSlc-nu/nu; obtained from Japan SLC) were intravenously injected in the tail vein with culture medium (200 μl) containing AUN ($15 \times 10^9$ CFU ml⁻¹ or $20 \times 10^9$ CFU ml⁻¹) 1 h after administration with saline (CMX; catalogue number 605000771, Chemix) (200 μl) containing heparin (5 mg per animal).

The anti-inflammatory agent dexamethasone was also used to clearly determine whether cytokine release syndrome caused by AUN could affect the viability of mice. AUN (200 μl, $15 \times 10^9$ CFU ml⁻¹) was intravenously injected into BALB/c-nu/nu mice (female; about 6 weeks old; average weight = 18 g; n = 5; genetic background, BALB/c; substrain, BALB/cCrSlc-nu/nu; obtained from Japan SLC) 24 h after administration of saline (200 μl) containing dexamethasone (catalogue number 041-18861, FUJIFILM Wako Pure Chemical) (2 mg per head).

To investigate in vivo anticancer therapy using human colorectal cancer, human ovarian cancer and human pancreatic cancer models, mice bearing tumours derived from HT29, SKOV3 or BxPC3 cells were generated by injecting 100 μl of the culture medium/Matrigel mixture (v/v = 1:1) containing $1 \times 10^6$ cells into the right side of the backs of the nude mice (female; 5 weeks old; average weight = 18 g; n = 5; genetic background, BALB/c; substrain, BALB/cCrSlc-nu/nu; obtained from Japan SLC). After approximately 20, 30 or 60 days (for HT29, SKOV3 or BxPC3 tumour cells, respectively) when the tumour

volumes reached ~400 mm$^3$ (HT29) or ~200 mm$^3$ (SKOV3 and BxPC3), the mice were intravenously injected with 200 µl of culture medium containing AUN ($1 \times 10^7$ CFU ml$^{-1}$ or $1 \times 10^9$ CFU ml$^{-1}$). A second dose of culture medium (200 µl) containing AUN ($15 \times 10^9$ CFU ml$^{-1}$ or $20 \times 10^9$ CFU ml$^{-1}$) was intravenously injected again 2 days after the first dose of AUN ($1 \times 10^7$ CFU ml$^{-1}$). The control experiments were also performed by i.v. injection of PBS (200 µl) into various types of human cancer model. The tumour volume and overall health (viability and body weight) were investigated, similarly to the anticancer study of Colon26 tumour-bearing BALB/c-nu/nu mice.

To generate a humanized mouse model, the right side of the backs of SCID mice (female; 5 weeks old; average weight = 18 g; $n = 5$; genetic background, C.B-17/Icr; substrain, C.B-17/Icr-scid/scidJcl; obtained from CLEA Japan) or NOD-SCID mice (female; 5 weeks old; average weight = 18 g; $n = 5$; genetic background, NOD/ShiJic; substrain, NOD/ShiJic-scid; obtained from CLEA Japan) was injected with 100 µl of the culture medium/Matrigel mixture (v/v = 1:1) containing $1 \times 10^6$ of Colon26 cells into the right side of the backs of the mice. Then, 10 days after Colon26 cell implantation, 200 µl of culture medium containing AUN ($1 \times 10^7$ CFU ml$^{-1}$ for SCID and $1 \times 10^8$ CFU ml$^{-1}$ for NOD-SCID) was intravenously injected. The first dose of culture medium (200 µl) containing AUN ($1 \times 10^7$ CFU ml$^{-1}$ for SCID and $1 \times 10^8$ CFU ml$^{-1}$ for NOD-SCID) was intravenously injected. After 2 days, the second dose of culture medium (200 µl) containing AUN ($7 \times 10^9$ CFU ml$^{-1}$ for SCID and $15 \times 10^9$ CFU ml$^{-1}$ for NOD-SCID) was intravenously injected. Antitumour efficacy tests of the single dose of AUN were also performed in Colon26-bearing SCID and NOD-SCID mice by administering AUN at a different concentration ($4.5 \times 10^9$ CFU ml$^{-1}$ for SCID and $3.0 \times 10^9$ CFU ml$^{-1}$ for NOD-SCID). Control experiments were also performed using PBS (200 µl).

To investigate the efficacy of AUN against an orthotopic human pancreatic cancer model, 50 µl of the culture medium containing $5 \times 10^6$ cells of BxPC3 was surgically inoculated into the pancreases of the mice (male; 8 weeks old; average weight = 30 g; $n = 5$; genetic background, BALB/c; substrain, BALB/cAJcl-nu/nu; obtained from CLEA Japan). Orthotopic human pancreatic cancer model was prepared by UNITECH. After 40 days, the mice were intravenously injected with 200 µl culture medium containing AUN ($1 \times 10^9$ CFU ml$^{-1}$ for the double-dose group or $7.8 \times 10^9$ CFU ml$^{-1}$ for the single-dose group). After 2 days, mice in the double-dose group were intravenously injected with another dose of culture medium (200 µl) containing AUN ($20 \times 10^9$ CFU ml$^{-1}$). Animals were euthanized on day 10, and their pancreases were collected. The pancreas weight of each mouse treated with PBS or AUN was measured on the same date. A control experiment without any treatments (that is, without inoculations of pancreas cancer cells or bacteria) was performed also.

The antiplatelet drugs were used to identify whether PLT affect the antitumour efficacy and thrombosis caused by AUN. AUN (200 µl, $7.8 \times 10^9$ CFU ml$^{-1}$) was intravenously injected into Colon26 tumour-bearing BALB/c-nu/nu mice (female; about 8 weeks old; average weight = 20 g; average tumour size ~200 mm$^3$; $n = 5$; genetic background, BALB/c; substrain, BALB/cCrSlc-nu/nu; obtained from Japan SLC) after administration of saline (200 µl) containing cilostazol (catalogue number 038-20661, FUJIFILM Wako Pure Chemical) (1 mg per head), acetylsalicylic acid (catalogue number 015-10262, FUJIFILM Wako Pure Chemical) (1 mg per head), ticlopidine hydrochloride (catalogue number 208-13971, FUJIFILM Wako Pure Chemical) (1 mg per head) or the mixture of cilostazol, acetylsalicylic acid and ticlopidine hydrochloride (each drug dose = 1 mg per head; total drug dose = 3 mg per head). AUN was injected 24 h after subcutaneous administration of cilostazol or the mixture of three antiplatelet drugs. Besides, AUN was administered 1 h after i.v. injection of acetylsalicylic acid or ticlopidine hydrochloride. Meanwhile, AUN (200 µl, $7.8 \times 10^9$ CFU ml$^{-1}$) was intravenously injected into Colon26 tumour-bearing BALB/c-nu/nu mice (female; about 8 weeks old; average weight = 20 g; average tumour

size ~200 mm$^3$; $n = 5$; genetic background, BALB/c; substrain, BALB/cCrSlc-nu/nu; obtained from Japan SLC) 1 h after i.v. administration with saline (200 µl) containing heparin (5 mg per animal) 24 h after subcutaneous administration of saline (200 µl) containing the mixture of cilostazol, acetylsalicylic acid and ticlopidine hydrochloride (each drug 1 mg per head; total drug dose = 3 mg per head).

Anti-Ly6G antibody was used to identify whether neutrophils affect the survival of mice. AUN (200 µl, $15 \times 10^9$ CFU ml$^{-1}$) was intravenously injected into BALB/c-nu/nu mice (female; about 6 weeks old; average weight = 18 g; $n = 5$; genetic background, BALB/c; substrain, BALB/cCrSlc-nu/nu; obtained from Japan SLC) 24 h after intraperitoneal administration of saline (200 µl) containing heparin sodium salt (5 mg per head). To study the effect of neutrophil on tumour efficacy, AUN (200 µl, $3.0 \times 10^9$ CFU ml$^{-1}$) was intravenously injected into Colon26 tumour-bearing BALB/c-nu/nu mice (female; about 8 weeks old; average weight = 20 g; average tumour size ~200 mm$^3$; $n = 5$; genetic background, BALB/c; substrain, BALB/cCrSlc-nu/nu; obtained from Japan SLC) 1 h after i.v. administration with saline (200 µl) containing heparin (5 mg per animal) 24 h after intraperitoneal administration of saline (200 µl) of anti-Ly6G antibody (anti-mouse Ly6G-InVivo; catalogue number A2158, Selleck) (0.2 mg per head) and subcutaneous administration of saline (200 µl) containing the mixture of cilostazol, acetylsalicylic acid and ticlopidine hydrochloride (each drug 1 mg per head; total drug dose = 3 mg per head).

The antibiotics were used to identify whether it could control the viability of AUN in mice. Saline (200 µl) containing imipenem (1 mg per head) was intraperitoneally injected into Colon26 tumour-bearing BALB/c-nu/nu mice (female; about 8 weeks old; average weight = 20 g; average tumour size ~200 mm$^3$; $n = 5$; genetic background, BALB/c; substrain, BALB/cCrSlc-nu/nu; obtained from Japan SLC) 24 h after i.v. administration of AUN (200 µl, $5 \times 10^9$ CFU ml$^{-1}$). After 24 h, the organs and tumours were carefully excised and weighed. After homogenizing thoroughly with a homogenizer pestle in 1 ml of PBS solution at 4 °C, the mixture was shaken for 20 min at a speed of $10 \times g$ at 15 °C. The supernatant was diluted 0, 10, 100 and 1,000 times with PBS, and then a sample (5 µl) was inoculated onto an agar plate. Finally, formed bacterial colonies were manually counted.

Intratumoural blood vessels were observed by using an angioscope (Bscan-ZD; GOKO) and software (GOKO Measure Plus V2.0.0; GOKO).

## Statistics and reproducibility

All experiments except those shown in Supplementary Information were performed in triplicate and repeated three or more times. Quantitative values are expressed as the mean ± s.e.m. of at least three independent experiments. Statistical differences were identified by Student's two-sided $t$-test, one-way or two-way analysis of variance, or log-rank (Mantel–Cox) test using GraphPad Prism, version 9.4.0 (GraphPad Software) and Microsoft Excel, version 2016 (Microsoft). A $P$ value of less than 0.05 was considered statistically significant. No covariates were tested. Normality of data distribution was not formally tested, but parametric tests were used based on standard assumptions. No correction for multiple comparisons was applied. Bayesian methods were not used, and no priors or Markov chain Monte Carlo settings were applied. The study did not involve hierarchical or complex designs. Effect size estimates (for example, Cohen's $d$ and Pearson's $r$) were not calculated. No data were excluded from the analyses.

## Reporting summary

Further information on research design is available in the Nature Portfolio Reporting Summary linked to this article.

## Data availability

The main data supporting the results in this study are available within the paper and its Supplementary Information. The raw and analysed

datasets from the study are too large to be publicly shared, but they are available for research purposes from the corresponding author on reasonable request. Source data are provided with this paper.

## Code availability

This study does not involve any custom code.

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

## Acknowledgements

This work was financially supported by Japan Society for the Promotion of Science (JSPS) KAKENHI Grant-in-Aid for Scientific Research (A) (grant number 23H00551), JSPS KAKENHI Grant-in-Aid for Challenging Research (Pioneering) (grant number 22K18440 and 25K21827), the Japan Science and Technology Agency (JST) for Adaptable and Seamless Technology Transfer Program through Target-driven R&D (grant number JPMJTR22U1), JST Program for co-creating start-up ecosystem (grant number JPMJSF2318), Institute for Fermentation, Osaka (IFO), the Uehara Memorial Foundation and Daiichi Sankyo Co., Ltd. S.I. thanks to JST SPRING (grant number JPMJSP2102). We also thank M. Kawahara (JAIST) and M. Deguchi (JAIST) for their dedicated support for the animal and cellular experiments.

## Author contributions

E.M. conceived and designed the study, supervised the project and wrote the initial paper. S.I. led and performed the majority of the experiments. T.N. conducted cytokine assays with and without heparin or dexamethasone premedication. M.S. performed qPCR analyses to determine bacterial ratios. Y.D. conducted comparative genome and transcriptome analyses of the bacteria, supported by N.T. H.N. validated the in vivo antitumour efficacy, with support from Y.O. and K.F. All authors (E.M., S.I., T.N., M.S., Y.D., N.T., H.N., Y.O. and K.F.) contributed to data analysis, discussed the results, participated in paper editing and approved the final version of the paper.

## Competing interests

E.M. is inventor on a patent application (WO2023223869A1) covering the anticancerous intratumoural bacteria. Y.O., H.N. and K.F. are the employees of Daiichi Sankyo Company, Limited. Y.O., H.N. and K.F. are shareholders of Daiichi Sankyo Company, Limited. This study was conducted as part of a collaborative research project with Daiichi Sankyo Company, Limited, which provided experimental and analytical support. However, the company was not involved in the interpretation of results or in the decision to publish. The other authors declare no competing interests.

## Additional information

**Correspondence and requests for materials** should be addressed to Eijiro Miyako.

# Reporting Summary

## Statistics

For all statistical analyses, confirm that the following items are present in the figure legend, table legend, main text, or Methods section.

| n/a | Confirmed | |
|---|---|---|
| ☐ | ☒ | The exact sample size (*n*) for each experimental group/condition, given as a discrete number and unit of measurement |
| ☐ | ☒ | A statement on whether measurements were taken from distinct samples or whether the same sample was measured repeatedly |
| ☐ | ☒ | The statistical test(s) used AND whether they are one- or two-sided<br>*Only common tests should be described solely by name; describe more complex techniques in the Methods section.* |
| ☒ | ☐ | A description of all covariates tested |
| ☒ | ☐ | A description of any assumptions or corrections, such as tests of normality and adjustment for multiple comparisons |
| ☐ | ☒ | A full description of the statistical parameters including central tendency (e.g. means) or other basic estimates (e.g. regression coefficient) AND variation (e.g. standard deviation) or associated estimates of uncertainty (e.g. confidence intervals) |
| ☐ | ☒ | For null hypothesis testing, the test statistic (e.g. *F*, *t*, *r*) with confidence intervals, effect sizes, degrees of freedom and *P* value noted<br>*Give P values as exact values whenever suitable.* |
| ☒ | ☐ | For Bayesian analysis, information on the choice of priors and Markov chain Monte Carlo settings |
| ☒ | ☐ | For hierarchical and complex designs, identification of the appropriate level for tests and full reporting of outcomes |
| ☒ | ☐ | Estimates of effect sizes (e.g. Cohen's *d*, Pearson's *r*), indicating how they were calculated |

*Our web collection on statistics for biologists contains articles on many of the points above.*

## Software and code

Policy information about availability of computer code

| | |
|---|---|
| Data collection | BZ-X800 Analyzer V1.1.2.4; CellSens V3.1; GOKO Measure Plus V2.0.0; MACSQuantify V2.13.3 |
| Data analysis | Excel 2016; GraphPad Prism 9.4.0; MACSQuantify V2.13.3; BZ-X800 Analyzer Ver1.1.2.4 |

For manuscripts utilizing custom algorithms or software that are central to the research but not yet described in published literature, software must be made available to editors and reviewers. We strongly encourage code deposition in a community repository (e.g. GitHub). See the Nature Portfolio guidelines for submitting code & software for further information.

## Data

Policy information about availability of data

All manuscripts must include a data availability statement. This statement should provide the following information, where applicable:
- Accession codes, unique identifiers, or web links for publicly available datasets
- A description of any restrictions on data availability
- For clinical datasets or third party data, please ensure that the statement adheres to our policy

The main data supporting the results in this study are available within the paper and its Supplementary Information. The raw and analysed datasets from the study are too large to be publicly shared, but they are available for research purposes from the corresponding author on reasonable request. Source data are provided with this paper.

## Research involving human participants, their data, or biological material

Policy information about studies with human participants or human data. See also policy information about sex, gender (identity/presentation), and sexual orientation and race, ethnicity and racism.

| | |
|---|---|
| Reporting on sex and gender | The donor was male. Sex was not a variable in the study design or analysis. |
| Reporting on race, ethnicity, or other socially relevant groupings | The donor was reported by the supplier as Black and Non-Hispanic. Race and ethnicity were not variables in the study design or analysis. |
| Population characteristics | The study used commercially available refrigerated human blood obtained from a single healthy donor, aged 32 years, identified as male, Black, and Non-Hispanic. |
| Recruitment | No direct recruitment was performed. The blood sample was purchased from a commercial supplier [BioIVT (College Park, MD, USA)], and donor information was limited to age, sex, and self-reported race/ethnicity. |
| Ethics oversight | Ethical approval was not required as the study used anonymized, commercially obtained human biological material with no identifiable personal information. |

Note that full information on the approval of the study protocol must also be provided in the manuscript.

# Field-specific reporting

Please select the one below that is the best fit for your research. If you are not sure, read the appropriate sections before making your selection.

☒ Life sciences    ☐ Behavioural & social sciences    ☐ Ecological, evolutionary & environmental sciences

For a reference copy of the document with all sections, see nature.com/documents/nr-reporting-summary-flat.pdf

# Life sciences study design

All studies must disclose on these points even when the disclosure is negative.

| | |
|---|---|
| Sample size | For experiments involving cell-free experiments, in vitro cell experiments, and phenotype analysis of the cells in tissues, n = 3 was chosen as the minimal replicate numbers. For in vivo antitumour experiments, n = 5 was chosen as the minimal replicate numbers. Sample sizes in the animal studies were determined on the basis of previous experimental experience (Xi Yang et al. Nano Today 37, 101100 (2021); Sheethal Reghu et al. Nano Today 52, 101966 (2023); Yun Qi et al. Advanced Functional Materials 34, 2305886 (2023). ) |
| Data exclusions | No data was excluded from this study |
| Replication | Experiments were repeated at least three independent experiments with similar results. All experiments were reproduced to reliably support conclusions stated in the manuscript. |
| Randomization | For the in vivo studies, the animals were randomly assigned to treatment groups at the outset of the study. Mice were assigned consecutive numbers across cages, and the groups were clustered consecutive numbers at random. All animals were sex- and age-matched within each experimental group. Different experiments used animals of different ages as appropriate. Male mice were used only in the BxPC3 orthotopic pancreatic tumour model; all other experiments employed female mice. No animals were excluded based on sex or age, and no selection bias was introduced in assigning animals to treatment groups. |
| Blinding | Investigators were blinded to group allocation during the experiments. |

# Reporting for specific materials, systems and methods

We require information from authors about some types of materials, experimental systems and methods used in many studies. Here, indicate whether each material, system or method listed is relevant to your study. If you are not sure if a list item applies to your research, read the appropriate section before selecting a response.

## Materials & experimental systems

| n/a | Involved in the study |
|---|---|
| ☐ | ☒ Antibodies |
| ☐ | ☒ Eukaryotic cell lines |
| ☒ | ☐ Palaeontology and archaeology |
| ☐ | ☒ Animals and other organisms |
| ☒ | ☐ Clinical data |
| ☒ | ☐ Dual use research of concern |
| ☒ | ☐ Plants |

## Methods

| n/a | Involved in the study |
|---|---|
| ☒ | ☐ ChIP-seq |
| ☐ | ☒ Flow cytometry |
| ☒ | ☐ MRI-based neuroimaging |

# Antibodies

| | |
|---|---|
| Antibodies used | Antibodies against F4/80 (mouse monoclonal, T-2028, BMA Biomedicals, 1:50), CD3 (rabbit monoclonal, ab16669, Abcam, 1:100), CD19 (rabit polyclonal, bs-0079R, Bioss, 1:100), CXCR4 (goat polyclonal, ab1670, Abcam, 1:100), Nkp46 (rabbit polyclonal, DF7599, Affinity Biosciences, 1:100),  Caspase-3 (Rabbit polyclonal, 9661S, Cell Signaling Technology, 1:00), TNF-α (Rabbit polyclonal, ab6671, Abcam, 1:100), IFN-γ (rabbit polyclonal, ab9657, Abcam, 1:100), IL-6 (rabbit polyclonal, bs-0782R, Bioss, 1:100), IL-1β (rabbit polyclonal, GTX100793, GenTex, 1:100), Fibrin (mouse monoclonal, MABS2155-25UG, Merck, 1:50), Transferrin (rabbit polyclonal, 17435-1-AP, Proteintech, 1:100), Ferritin (rabbit polyclonal, 11682-1-AP, Proteintech, 1:100), CA19-9 (mouse monoclonal, ab289665, Abcam, 1:500), and digoxigenin-peroxidase (sheep polyclonal, S7100, Merck Millipore, non-dilution) were used for IHC staining. Antibodies against CD3 (FITC-labelled, Human cell line monoclonal, 130119-798, Miltenyi Biotech), CD335 (NKp46) (PE-labelled, Human cell line monoclonal, 130-112-358, Miltenyi Biotech), CD45R (B220) (PerCP-Vio 700-labelled, Human cell line monoclonal, 130-110-850, Miltenyi Biotech), Ly-6G (PE-Vio 770-labelled, Human cell line monoclonal, 130-121-438, Miltenyi Biotech), F4/80 (APC-labelled, Human cell line monoclonal, 130-116-525, Miltenyi Biotech), and CD45 (APC-Vio 770-labelled, Human cell line monoclonal, 130-110-800, Miltenyi Biotech) were used for flow cytometry. |
| Validation | All antibodies were used in the study according to the profile of manufacturers. Antibody validation for preparation of immunohistochemistry or flowcytometry was validated by the supplier and confirmed in Figure 3 and Supplementary Figures 9, 26, 42, and 46. |

# Eukaryotic cell lines

Policy information about cell lines and Sex and Gender in Research

| | |
|---|---|
| Cell line source(s) | The Colon26 cell line (catalogue no. RBC2657) was obtained from RIKEN BioResource Research Center (Ibaraki, Japan). Human normal diploid fibroblasts (MRC5) cell line (catalogue no. JCRB9008) was obtained from the Japanese Collection of Research Bioresources Cell Bank (Tokyo, Japan). Human pancreatic cancer (BxPC3) cell (catalogue no. CRL1687) was obtained from American Type Culture Collection (Manassas, VA, USA). Human ovarian cancer (SKOV3) (catalogue no. EC91091004-F0) and human colorectal adenocarcinoma (HT29) cells (catalogue no. EC91072201-G0) cells were obtained from KAC Co., Ltd. (Tokyo, Japan). |
| Authentication | These cell lines were authenticated by the supplier using STR analysis. |
| Mycoplasma contamination | No contamination was detected by the supplier using Hoechst DNA stain method, agar culture method, PCR-based assay. |
| Commonly misidentified lines (See ICLAC register) | These cell lines that we used were not listed in commonly misidentified lines in ICLAC Register. |

# Animals and other research organisms

Policy information about studies involving animals; ARRIVE guidelines recommended for reporting animal research, and Sex and Gender in Research

| | |
|---|---|
| Laboratory animals | Female BALB/c(4–8 weeks old) and Female BALB/c-nu/nu mice (4 and 5 weeks old) were purchased from Japan SLC (Hamamatsu, Japan). Female NOD-SCID (4 and 5 weeks old) and female SCID mice (4 and 5 weeks old) were obtained from Japan Clea (Tokyo, Japan). |
| Wild animals | The study did not involve wild animals. |
| Reporting on sex | The information was described in Methods in more details. |
| Field-collected samples | The study did not involve samples collected from the field. |
| Ethics oversight | The animal experiments were conducted following the protocols approved by the Institutional Animal Care and Use Committee of Japan Advanced Institute of Science and Technology (No. 07-001). Mice were group-housed in ventilated clear plastic cages under appropriate ambient temperature (~23°C), humidity (~50%), and standard 12 h:12 h light:dark conditions. Experimental group sizes were approved by the regulatory authorities for animal welfare after being defined to balance statistical power, feasibility, and |

ethical aspects. Maximal tumour burden permitted is 3000 mm3, and the maximal tumour size/burden was not exceeded in the experiments.

Note that full information on the approval of the study protocol must also be provided in the manuscript.

# Plants

| Seed stocks | Irrelevant to experiments. |
|---|---|
| Novel plant genotypes | Irrelevant to experiments. |
| Authentication | Irrelevant to experiments. |

# Flow Cytometry

## Plots

Confirm that:

☒ The axis labels state the marker and fluorochrome used (e.g. CD4-FITC).

☒ The axis scales are clearly visible. Include numbers along axes only for bottom left plot of group (a 'group' is an analysis of identical markers).

☒ All plots are contour plots with outliers or pseudocolor plots.

☒ A numerical value for number of cells or percentage (with statistics) is provided.

## Methodology

| Sample preparation | We have described in Methods in more details. |
|---|---|
| Instrument | MACSQuant® Analyzer 16 (Miltenyi Biotec, Bergisch Gladbach, Germany) |
| Software | MACSQuantify V2.13.3 (Miltenyi Biotec) |
| Cell population abundance | At least 10,000 relevant events were acquired for all FACS analysis. |
| Gating strategy | In general, cells were first gated on FSC/SSC. Singlet cells were gated using FSC-H and FSC-A. Dead cells were then excluded and further surface and intracellular antigen gating was performed on the live cell population. |

☒ Tick this box to confirm that a figure exemplifying the gating strategy is provided in the Supplementary Information.

