## [Peer Review File · Nature Biomedical Engineering]

Selective intratumoural thrombosis and necrosis induced by tumour-resident oncolytic bacteria lead to potent anticancer effects

Corresponding Author: Prof Eijiro Miyako

Version 0:

Decision Letter:

Dear Professor Miyako,

Thank you for submitting to *Nature Biomedical Engineering* your Presubmission Enquiry, "Transcendent oncolytic ability of bacterial consortium AUN in immunocompromised models".

As you may know, we screen Presubmission Enquiries against our editorial criteria. These editorial judgements are based on considerations of fit to the journal's scope and, when enough information is provided, of the degree of advance, broad implications, and breadth and depth of the work.

I am sorry for not being able to look at your study earlier. The topic of the Presubmission Enquiry is within the remit of the journal, and we would like to invite you to submit a full manuscript so that we can carry out a full editorial assessment.

I should also ask you to please fill in our [reporting summary](https://www.nature.com/authors/policies/ReportingSummary.pdf) and [policy checklist](https://www.nature.com/authors/policies/Policy.pdf). (Please note that these forms are dynamic PDF files that can only be properly visualized and filled in by using [Acrobat Reader](https://get.adobe.com/reader).)

Both documents are aimed at ensuring good reporting standards and at easing the interpretation of results, and will be available to any reviewers. Should the manuscript be eventually published, the reporting summary will be attached to the published PDF of the paper and will also be available as supplementary information. More information is available on the [editorial policies](http://www.nature.com/authors/policies/availability.html#requirements) page.

Moreover, we highly recommend that you use our [manuscript template](https://bit.ly/3FoGyHd). This will help you ensure that the manuscript complies with our data-presentation recommendations, that it includes all the necessary sections, and that it is structured to facilitate the assessment of peer reviewers and editors. In particular, please make sure that the manuscript provides thorough information on statistics and methods, and that the images comply with our [guidelines for image integrity](https://www.nature.com/nature-portfolio/editorial-policies/image-integrity).

When you are ready to submit the manuscript, please upload the manuscript files as well as the reporting summary and policy checklist. DELETE_IF_COVER_LETTER_ALREADY_PROVIDED Please also upload a brief cover letter that describes the main results of the work and places it into a broader context.

Best wishes,
Filipe

Dr Filipe Almeida
Senior Editor, [Nature Biomedical Engineering](http://www.nature.com/nbme)

Version 1:

Decision Letter:

Dear Dr. Miyako,

Thank you again for submitting to *Nature Biomedical Engineering* your manuscript, "Oncolytic ability of bacterial consortium AUN in immunocompromised models". The manuscript has been seen by 3 experts, whose reports you will find at the end of this message.

You will see that the reviewers appreciate the work. However, they express concerns about the degree of support for the claims, and provide useful suggestions for improvement. We hope that with substantial further work you can address the criticisms and convince the reviewers of the merits of the study. In particular, we would expect that a revised version of the manuscript provides:

- * Discussion on the translational and clinical feasibility of the tumor-resident bacterial consortium, addressing potential limitations, such as the challenge of maintaining precise bacterial ratios and its implications for therapeutic application, as highlighted by Reviewer #2.
- * Examination of the tumor-resident bacterial consortium's specificity, as suggested by Reviewer #2.
- * Deeper mechanistic insights, as suggested by Reviewer #2 and Reviewer #3.
- * Additional evidence of therapeutic effects in preclinical models that more closely mimic human pathology, as highlighted by Reviewer #2.

When you are ready to resubmit your manuscript, please upload the revised files, a point-by-point rebuttal to the comments from all reviewers, the [reporting summary](https://www.nature.com/authors/policies/ReportingSummary.pdf), and a cover letter that explains the main improvements included in the revision and responds to any points highlighted in this decision.

Please follow the following recommendations:

- * Clearly highlight any amendments to the text and figures to help the reviewers and editors find and understand the changes (yet keep in mind that excessive marking can hinder readability).
- * If you and your co-authors disagree with a criticism, provide the arguments to the reviewer (optionally, indicate the relevant points in the cover letter).
- * If a criticism or suggestion is not addressed, please indicate so in the rebuttal to the reviewer comments and explain the reason(s).
- * Consider including responses to any criticisms raised by more than one reviewer at the beginning of the rebuttal, in a section addressed to all reviewers.
- * The rebuttal should include the reviewer comments in point-by-point format (please note that we provide all reviewers will the reports as they appear at the end of this message).
- * Provide the rebuttal to the reviewer comments and the cover letter as separate files.

We expect that you will be able to resubmit the manuscript within 20 weeks of receiving this message. If this is the case, you will be protected against potential scooping. Otherwise, we will be happy to consider a revised manuscript as long as the significance of the work is not compromised by work published elsewhere or accepted for publication at *Nature Biomedical Engineering*.

We hope that you will find the referee reports helpful when revising the work. Please do not hesitate to contact me should you have any questions.

Best wishes,

Filipe

Dr Filipe Almeida
Senior Editor, <http://www.nature.com/nbme> > *Nature Biomedical Engineering*

Reviewer #1 (Report for the authors (Required)):

The authors have developed an intratumoral bacterial consortium that demonstrates selective growth and proliferation within

a targeted tumor microenvironment. This study originated from intriguing concepts and presents potential therapeutic advantages for immunosuppressive tumors. However, there are several concerns that need to be addressed before publishable.

1. What's the function of the *Rhodopseudomonas palustris*? Please elaborate in detail in the introduction.
 - 1.1. The authors mentioned that "Even after repeated subculturing the ratio of AUN was consistently A-gyo:UN-gyo = 3:97." The authors should do more investigation to explain why they choose this ratio. Following intravenous injection, can the two bacteria sustain this ratio within the tumor?
 - 1.2. Do the *Rhodopseudomonas palustris* directly contribute to the synthesis of fibers? What is the composition of the bacterial fiber?
2. The pharmacokinetics of both bacteria should be investigated separately.
3. The authors showed that "Cancerous tumors typically use iron for cancer initiation, tumor growth, and metastasis. We speculated that AUN might also absorb and degrade hemoglobin from blood cells in the tumor microenvironment, leading to tumor-specific hemolysis and thrombosis." The authors should conduct more investigation to substantiate their claim.
4. In an antitumor study, authors should compare with the mono-bacteria.
5. The authors mentioned that "Although we previously confirmed that various immune cells assisted anticancer efficacy of AUN in tumor-bearing immunocompetent mice, these results clearly indicate that the expression of immune cells had no crucial role in the strong antitumor efficacy of AUN." It is not objective to draw these conclusions in an immunosuppressed mouse model.
6. There are some spelling mistakes in the manuscript, please double check carefully.

Reviewer #2 (Report for the authors (Required)):

This manuscript explores the therapeutic potential of the tumor-resident bacterial consortium AUN in immunocompromised cancer models, emphasizing its unique mechanisms of tumor-specific thrombosis and oncolysis. While the study provides significant insights, several points require clarification and further discussion to enhance its scientific impact:

1. A-gyo:UN-gyo Ratio

The manuscript emphasizes the 3:97 ratio of A-gyo to UN-gyo but does not explain its determination or criticality to efficacy. Experimental validation of how variations in this ratio affect efficacy, stability, and safety would add significant value.

2. Mechanisms of Thrombosis

The discussion should explore the multiple potential causes of AUN-induced thrombosis in more depth, including vascular invasion, toxin- and enzyme-induced damage, altered iron metabolism, and platelet-independent fibrinogen activation. Understanding the interplay of these mechanisms with tumor-specific thrombosis is essential.

3. Hemolysis Specificity

Clarification is needed on whether AUN-induced hemolysis is confined to tumor vasculature or affects normal red blood cells. Data suggesting possible effects on normal vasculature (e.g., Figure 4i) raise safety concerns for human application.

4. Cytokine Effects and Tumor Necrosis

Further insights into the roles of TNF- α , IFN- γ , and other cytokines in vascular damage and necrosis across tumor models would strengthen the mechanistic framework. Comparing cytokine distribution and heterogeneity in tumor tissues could illuminate differences in therapeutic response.

5. Translatability and Safety

The manuscript underestimates potential human risks due to physiological differences between mice and humans. Mice lacking baseline atherosclerosis and less prone to thrombosis may not adequately model human vascular risks. Incorporating models that better mimic human pathology, such as hyperlipidemic or atherosclerosis-prone models, is essential for assessing safety.

6. General Observations

Differences in tumor responses (e.g., hemorrhagic necrosis in Colon26 vs. gradual damage in BxPC3) suggest variability that warrants further exploration.

The reason for using both PBS and Non-treatment controls should be explicitly discussed to explain their respective roles.

7. Figure 3C: TNF- α and fibrinogen staining require more interpretation, particularly fibrinogen's biological significance in thrombosis and inflammation. Regarding thrombosis, fibrin should be examined rather than fibrinogen.

8. Figure 4J: The dynamics presented are unclear; a video-based explanation may enhance clarity.

9. Figure 5: The distinction between PBS and Non-treatment controls needs a more precise definition.

10. Supplementary Figure S2: The bar graph colors should match the legend order.

Reviewer #3 (Report for the authors (Required)):

Tumor-resident bacteria are a highly promising form of cancer therapy. This manuscript designed a bacterial consortium AUN composed of *Proteus mirabilis* and *Rhodopseudomonas palustris* and investigated the microbial characteristics and potential antitumor efficacy of AUN. The AUN demonstrated tumor-suppressive effects across multiple immunodeficient models. Furthermore, through integrated multi-dimensional analysis combining genomics, transcriptomics, histopathology, and in vivo imaging techniques, the study revealed AUN's tumor-targeting mechanisms mediated by thrombosis induction, bacterial oncolysis, and unique metabolic regulation. In conclusion, the design of the manuscript is innovative and offers a potential new strategy for treating tumors in immunosuppressed patients, but several key issues still need to be clarified. Major comments:

1. The manuscript mentions that high-dose AUN administration caused mortality in mice, although this issue was mitigated through fractionated dosing. However, this observation suggests a narrow therapeutic window for AUN, necessitating cautious clinical implementation. It is recommended to explore either lower effective doses or combination therapy strategies to optimize safety and efficacy.
2. The manuscript's biosafety data focus on short-term side effects but lack long-term observations. The long-term safety of AUN and potential drug resistance issues require in-depth discussion.
3. Although the mechanisms section is detailed, some conclusions may require more direct experimental evidence. For example, the manuscript mentions, "The exhaustion of iron in the tumor milieu by the increased iron requirements of AUN via bacterial cross-talking (as indicated by the transcriptome analysis) might also indirectly be involved in the tumor suppression." but there are no data to demonstrate the relationship between iron deprivation and tumor suppression directly.
4. It was proposed in the manuscript that AUN induces thrombosis through activation of platelets (PLT) and fibrinogen, but experiments showed that antiplatelet drugs failed to inhibit their effects (Figure S21). This suggests that thrombosis may depend on other pathways (e.g., direct damage to the vascular endothelium by bacteria or release of procoagulant factors). Have specific procoagulant molecules secreted by AUN (e.g., thrombin-like enzymes) been detected? Is there evidence that AUN alters the coagulation status of the tumor microenvironment through metabolism, e.g., iron uptake?
5. A-gyo accounted for only 3% of AUN (97% of UN-gyo), but the gene expression profiles of the consortium were significantly different from those of the single strains (Fig. 1e-f). How do the two regulate thrombosis and tumor lysis through cross-talk (cross-talk)? Are there co-culture experiments to demonstrate their synergistic effects?

Minor comments:

1. In the manuscript, the authors refer to "AUN are beyond species difference, being highly effective as a universal therapeutic agent in various immunocompromised models derived from both human and murine cancer cells." However, validating the tumor effect of AUN in the manuscript focuses on colon and pancreatic cancer models. There is limited data on other tumor types, and the authors must correct this description.
2. BALB/c-nu/nu, SCID, and NOD-SCID models were used in the manuscript, but these models still retain some innate immunity, e.g., residual NK cells in NOD-SCID). Did the authors test the role of innate immune cells, such as neutrophils and macrophages, in thrombosis? In addition, patients with clinical immunosuppression are often accompanied by complex immune dysregulation due to chemotherapy or radiotherapy. Consider whether the efficacy of AUN is stable in such scenarios.
3. Immunodeficient mouse models do not fully mimic clinically immunosuppressed patients, and the authors need to discuss the possible risk of immune clearance of AUN in humans.
4. Figure 3e demonstrates only the H&E staining of the thrombus and can be supplemented with other characterization evidence to enhance the reliability of the conclusions.
5. Figure 4 has a low resolution for b and f. The authors need to provide clearer optical microscopic images.
6. Although AUN is described as 'non-pathogenic', A-gyo carries several hemolysin and toxin genes (Figure 4h). Has its long-term toxicity to normal tissues (e.g. liver, kidney) been assessed? Can the toxicity be further reduced by gene editing?
7. The authors need to provide data on the hemolytic activity of AUN in human cell lines to support clinical safety.

Version 2:

Decision Letter:

Dear Prof Miyako,

Thank you for your revised manuscript, "Transcendent oncolytic ability of bacterial consortium AUN in immunocompromised models". Having consulted with the original reviewers, I am pleased to write that we shall be happy to publish the manuscript in *Nature Biomedical Engineering*.

We will be performing detailed checks on your manuscript, and in due course will send you a checklist detailing our editorial and formatting requirements. You will need to follow these instructions before you upload the final manuscript files.

Best wishes,
Filipe

Dr Filipe Almeida
Senior Editor, Nature Biomedical Engineering

Reviewer #1 (Report for the authors (Required)):

The revised manuscript has been carefully re-evaluated. It is evident that the authors have made considerable efforts to address all the concerns raised in the previous review. The revisions have significantly enhanced the quality and clarity of the manuscript.

The manuscript now aligns well with the standards of the journal and makes a valuable contribution to the field. Thus, it has been decided to accept the manuscript for publication.

Reviewer #2 (Report for the authors (Required)):

The authors' responses and revisions substantially address the issues I pointed out, and the major concerns have been adequately resolved.

Reviewer #3 (Report for the authors (Required)):

The revised manuscript is ready for publication.

Version 3:

Decision Letter:

Dear Prof Miyako,

I am happy to inform you that your manuscript, "Selective intratumoural thrombosis and necrosis induced by tumour-resident oncolytic bacteria lead to potent anticancer effects", has now been accepted for publication in *Nature Biomedical Engineering*.

Over the next few weeks, the figures will be checked for production quality, the text edited to ensure that it conforms to house style, and the manuscript typeset.

Our Articles are published about 40 days after the acceptance date (we recommend that you inform your institutional press office of this timeframe), and you will be notified of the actual publication date a few days in advance. Articles can be published any working day of the week, and are pushed live shortly after 10 am London time.

Publishing agreement. You will be asked to digitally sign a publishing agreement (grant of rights). After the signed publishing agreement has been received, the proofs of the article will be sent to you for review. If you have any queries during the production process, or you cannot meet the requested deadline for returning the proofs, please contact rjsproduction@springernature.com.

Nature Biomedical Engineering is a Transformative Journal. Authors may publish their research with us through the traditional subscription access route, or make their paper immediately open access through payment of an article-processing charge. More [information about publication options](https://www.springernature.com/gp/open-research/transformative-journals) is available.

You may need to take specific actions to [comply](https://www.springernature.com/gp/open-research/funding/policy-compliance-faqs) with funder and institutional open-access mandates. If the work described in the accepted manuscript is supported by a funder that requires immediate open access (as outlined, for example, by [Plan S](https://www.springernature.com/gp/open-research/plan-s-compliance)) and your manuscript was originally submitted on or after January 1st 2021, then you should select the gold OA route. Authors selecting subscription publication will need to accept our standard licensing terms (including our [self-archiving policies](https://www.springernature.com/gp/open-research/policies/journal-policies)), and these will supersede any other terms that the author or any third party may assert apply to any version of the manuscript.

Acceptance of your manuscript is conditional on agreement, by all authors, with both our [media embargo](http://www.nature.com/authors/policies/embargo.html) and [confidentiality and pre-publicity](http://www.nature.com/authors/policies/confidentiality.html) policies. In particular, you may arrange your own publicity of the Article (for instance, through your institutional press office), as long as you ensure that journalists strictly adhere to the media embargo.

To assist you in disseminating the work, as soon as the Article is published you will be able to take advantage of the Springer Nature [SharedIt](https://www.springernature.com/gp/researchers/sharedit) initiative to [generate a unique shareable link to the Article](http://authors.springernature.com/share) that will allow anyone (with or without a subscription) to read it. Recipients of the link who are subscribers will also be able to download and print

the PDF.

Thank you for having submitted this work to *Nature Biomedical Engineering*.

Best wishes,
Filipe

Dr Filipe Almeida
Senior Editor, <http://www.nature.com/nbme>>*Nature Biomedical Engineering*

Reviewer #1

The authors have developed an intratumoral bacterial consortium that demonstrates selective growth and proliferation within a targeted tumor microenvironment. This study originated from intriguing concepts and presents potential therapeutic advantages for immunosuppressive tumors. However, there are several concerns that need to be addressed before publishable.

General response: We would like to thank this referee for his/her beneficial and encouraging comments and a compliment on the work. According to valuable suggestions by the referee, we have carefully modified the manuscript.

Comment 1: What's the function of the *Rhodopseudomonas palustris*? Please elaborate in detail in the introduction.

Our Response 1: We have added the following description in the introduction according to the comment.

Page 3, Line 20; “We also found that the UN-gyo could empower the anticancer efficacy and safety of AUN via the stunning functions: (i) The suppression of biogenic activity (pathogenicity) of A-gyo, (ii) Increase of cancer-specific cytotoxicity by facilitation of fibrous structural transformation of A-gyo under the coexisting with cancer cells (oncometabolites), (iii) Attenuation of hemolysis activity of AUN, and (iv) Increase of iron requirements of AUN.”

Comment 2: The authors mentioned that “Even after repeated subculturing the ratio of AUN was consistently A-gyo:UN-gyo = 3:97.” The authors should do more investigation to explain why they choose this ratio. Following intravenous injection, can the two bacteria sustain this ratio within the tumor?

Our Response 2: We do not intentionally choose the bacterial ratio indeed because the ratio naturally gets to be A-gyo:UN-gyo \approx 3:97 by culturing. To clarify this more, we have precisely analyzed the bacterial ratios in culture media using qPCR for 9 days at gene level in addition to fluorescent microscopic investigations and colony assays. As the result, we confirm that the ratio of AUN is definitely A-gyo:UN-gyo \approx 3:97 even after repeated subculturing and cryopreservation (please see **Supplementary Figure S1**). Besides, when the isolated A-gyo and UN-gyo were intentionally mixed to A-gyo:UN-gyo = 3:97, we had observed that the anticancer efficacy of the mixture was decreased probably due to insufficient bacterial cross-talk without co-culturing [please read my previous paper; Goto et al. *Adv. Sci.* **10**, 23016 (2023)]. We also made sure that nature takes back to the original ratio of AUN (A-gyo:UN-gyo \approx 3:97) 7 days after culturing even when the ratio was intentionally adjusted as A-gyo:UN-gyo = 10:90, 50:50, or 90:10 before culturing (please see **Supplementary Figure S2**). We consider that these results potentially indicate that A-gyo has a symbiotic relationship (commensalism) with UN-gyo due to the Cys metabolism in A-gyo cell. Meanwhile, the artificial mixed ratios of AUN (A-gyo:UN-gyo = 15:85, 25:75, 50:50, or 97:3) caused mice to die within 48 h. Therefore, we believe that nature-prepared “golden” ratio of AUN (A-gyo:UN-gyo \approx 3:97) has both of strong anticancer efficacy and high safety. In any case, the following description has just been added in the main text.

Page 4, Line 13; “Maintaining precise bacterial ratios is significant for the translational and clinical feasibility of the tumor-resident bacterial consortium. Even after repeated subculturing, the ratio of AUN was consistently A-gyo:UN-gyo \approx 3:97, as confirmed using fluorescence microscopy and colony assays.²⁷ We have further analyzed the bacterial ratios in media using quantitative polymerase chain reaction (qPCR) (**Supplementary Figure S1 and S2**). As the result, the ratio of AUN is certainly A-gyo:UN-gyo \approx 3:97 even after repeated subculturing (**Supplementary Figure S1**). We also confirmed that the ratios get naturally back to be the original ratio of AUN (A-gyo:UN-gyo \approx 3:97) 7 days after culturing even when the ratios were intentionally prepared as A-gyo:UN-gyo = 10:90, 50:50, or 97:3 in advance (**Supplementary Figure S2**). These results potentially indicate that A-gyo has a symbiotic relationship (commensalism) with UN-gyo due to the Cys metabolism in A-gyo cell, as we previously reported.²⁷ Meanwhile, the artificial mixed ratios of AUN (A-gyo:UN-gyo = 15:85, 25:75, 50:50, or 97:3) caused Colon26-bearing BALB/c mice (N = 5 for each ratio) to die within 48 h. Therefore, we believe that nature-prepared “golden” ratio of AUN (A-gyo:UN-gyo \approx 3:97) has both of strong anticancer efficacy and high safety.”

Comment 3: Do the *Rhodopseudomonas palustris* directly contribute to the synthesis of fibers? What is the composition of the bacterial fiber?

Our Response 3: We consider that the composition of the bacterial fiber is only A-gyo. Indeed, we have confirmed that UN-gyo does not directly contribute as a component in the transformed fibrous structure of A-gyo by fluorescent microscopy (please see **Supplementary Figure S35**). The following description has also been added in the main text.

Page 12, Line 1; “We confirmed that UN-gyo does not directly contribute as a component on the transformation of fibrous structure of A-gyo by fluorescent microscopy (**Supplementary Figure S35**). In fact, the fluorescent pink dots derived from UN-gyo are not embedded in the A-gyo fibers.”

Comment 4: The pharmacokinetics of both bacteria should be investigated separately.

Our Response 4: We had already investigated the pharmacokinetics of A-gyo and UN-gyo, separately, and published the data in the different journals [Goto et al. *Adv. Sci.* **10**, 23016 (2023) and Yang et al. *Nano Today* **37**, 101100 (2021).] (please see the attached **Figure R1**). After the disappearance of tumors by the treatment of AUN, both bacterial colonies derived from A-gyo and UN-gyo are not detectable at all because there are no tumors, namely, comfortable residence for A-gyo and UN-gyo, anymore (the attached **Figure 1a**). The H&E histological analysis of tissues of the harvested vital organs, the heart, the liver, the lung, the kidney, and the spleen had also revealed no significant abnormalities or evidence of toxicity in the AUN and A-gyo treatments (the attached **Figure 1b**). The colony number of UN-gyo in an extracted tumor reached approximately 1×10^6 CFU at 168 h post-injection, whereas, the colonies of UN-gyo from vital organs were not completely detectable post-injection of UN-gyo at 168 h (the attached **Figure 1c and 1d**). We thus assume that most UN-gyo circulating through mice are eliminated by phagocytic immune cells, such as macrophages and neutrophils, at the first stage within 24 h. Surviving UN-gyo can proliferate in solid tumors, providing “safe” shelters for bacteria against aggressive immune system cells. In fact, cancer cells have immunosuppression mechanisms to evade attacking immune cells.

[Figure redacted]

Figure R1 for reviewing only. Biocompatibility of functional bacteria. (a) Numbers and images of the bacterial colony of organs in Colon-26-tumor-bearing mice after i.v. injection of A-gyo (left) (200 μ L, 1×10^8 CFU) or AUN (right) (200 μ L, 1×10^9 CFU) after 25 days. Data are represented as mean \pm SEM; n = 5 independent experiments. N. D., not detectable. (b) H&E staining in conventional organs sectioned after i.v. injection of A-gyo (left) (200 μ L, 1×10^8 CFU) or AUN (right) (200 μ L, 1×10^9 CFU) after 30 days. (c) Bacteria colony of organs/tumors of Colon26 tumor-bearing mice over time after intravenous injection of UN-gyo (200 μ L, 1×10^9 CFU) or PBS. (d) Colony number of Colon26 tumor-bearing mice after intravenous injection of UN-gyo (200 μ L, 1×10^9 CFU) at different time points. Data are represented as means \pm standard errors of the mean (SEM); n = 4 biologically independent organs and tumors. n.d., not detectable; *, $p < 0.05$; ***, $p < 0.001$.

Comment 5: The authors showed that “Cancerous tumors typically use iron for cancer initiation, tumor growth, and metastasis. We speculated that AUN might also absorb and degrade hemoglobin from blood cells in the tumor microenvironment, leading to tumor-specific hemolysis and thrombosis.” The authors should conduct more investigation to substantiate their claim.

Our Response 5: Firstly, the description has just been revised as follows in order to address the comment.

Page 4, Line 34; “Cancerous tumors typically use iron for cancer initiation, tumor growth, and metastasis.³² We speculated that AUN might also deplete iron from cancerous tumors and blood in the tumor microenvironment, leading to tumor-specific suppression (a mechanism described in detail in the section **Mechanism of tumor suppression**).^{33,34}”

Next, we have actually surveyed iron metabolism in the tumor microenvironment by IHC staining and qPCR analyses of transferrin and ferritin for supporting the aforesaid hypothesis (please see **Supplementary Figure S42**). The following description has also been added in the main text.

Page 13, Line 15; “Meanwhile, as we mentioned earlier, iron is essential for various cellular functions of cancer, and iron depletion strategies have a strong anti-proliferative effect on tumor cells.³²⁻³⁴ We found that transferrin and ferritin in the tumor microenvironment certainly were depleted by AUN treatment presumably owing to the increase of iron requirements of AUN via bacterial cross-talking (as indicated by the transcriptome analysis) (**Supplementary Figure S42**).”

Comment 6: In antitumor study, authors should compare with the mono-bacteria.

Our Response 6: We had already evaluated the antitumor activities of each mono-bacterium (A-gyo and UN-gyo) in addition to bacterial consortium AUN against various murine cancers-bearing immunocompetent mice [please read our previous literature for more details; Goto et al. *Adv. Sci.* **10**, 23016 (2023)]. As the results, AUN exhibits the strongest antitumor efficacy and safety in comparison with A-gyo and UN-gyo. Meanwhile, A-gyo and UN-gyo cannot express anticancer activities equivalent to AUN even in an immunological “hot” tumor microenvironment where is easier curable rather than a tumor in an immunocompromised mouse. Therefore, we applied only AUN for the current study for further investigation of its potential activity against immunocompromised models. Many thanks again for your understanding.

Comment 7: The authors mentioned that “Although we previously confirmed that various immune cells assisted anticancer efficacy of AUN in tumor-bearing immunocompetent mice, these results clearly indicate that the expression of immune cells had no crucial role in the strong antitumor efficacy of AUN.” It is not objective to draw these conclusions in an immunosuppressed mouse model.

Our Response 7: We agree with the comment of the reviewer. The description has been omitted to avoid misunderstanding of readers.

Comment 8: There are some spelling mistakes in the manuscript, please double check carefully.

Our Response 8: We have carefully amended the whole of manuscript for collection of typos. Many thanks again for your valuable comments and understanding.

Reviewer #2

This manuscript explores the therapeutic potential of the tumor-resident bacterial consortium AUN in immunocompromised cancer models, emphasizing its unique mechanisms of tumor-specific thrombosis and oncolysis. While the study provides significant insights, several points require clarification and further discussion to enhance its scientific impact.

General response: We really appreciate for the reviewer's beneficial and encouraging comments. we have carefully amended the manuscript in accordance with the reviewer's valuable comments. We are very glad that the revised version is sophisticated by the useful advises from the reviewer.

Comment 1: A-gyo:UN-gyo Ratio

The manuscript emphasizes the 3:97 ratios of A-gyo to UN-gyo but does not explain its determination or criticality to efficacy. Experimental validation of how variations in this ratio affect efficacy, stability, and safety would add significant value.

Our Response 1: We do not intentionally choose the bacterial ratio indeed because the ratio naturally gets to be A-gyo:UN-gyo \approx 3:97 by culturing. To clarify this more, we have precisely analyzed the bacterial ratios in culture media using qPCR for 9 days at gene level in addition to fluorescent microscopic investigations and colony assays. As the result, we confirm that the ratio of AUN is definitely A-gyo:UN-gyo \approx 3:97 even after repeated subculturing and cryopreservation (please see **Supplementary Figure S1**). Besides, when the isolated A-gyo and UN-gyo were intentionally mixed to A-gyo:UN-gyo = 3:97, we had observed that the anticancer efficacy of the mixture was decreased probably due to insufficient bacterial cross-talk without co-culturing (please read my previous paper; Goto et al. *Adv. Sci.* **10**, 23016 (2023)). We also made sure that nature takes back to the original ratio of AUN (A-gyo:UN-gyo \approx 3:97) 7 days after culturing even when the ratio was intentionally adjusted as A-gyo:UN-gyo = 10:90, 50:50, or 97:3 before culturing (please see **Supplementary Figure S2**). We consider that these results potentially indicate that A-gyo has a symbiotic relationship (commensalism) with UN-gyo due to the Cys metabolism in A-gyo cell. Meanwhile, the artificial mixed ratios of AUN (A-gyo:UN-gyo = 15:85, 25:75, 50:50, or 97:3) caused mice to die within 48 h. Therefore, we believe that nature-prepared "golden" ratio of AUN (A-gyo:UN-gyo \approx 3:97) has both of strong anticancer efficacy and high safety. In any case, the following description has just been added in the main text.

Page 4, Line 13; "Maintaining precise bacterial ratios is significant for the translational and clinical feasibility of the tumor-resident bacterial consortium. Even after repeated subculturing, the ratio of AUN was consistently

A-gyo:UN-gyo \approx 3:97, as confirmed using fluorescence microscopy and colony assays.²⁷ We have further analyzed the bacterial ratios in media using quantitative polymerase chain reaction (qPCR) (**Supplementary Figure S1 and S2**). As the result, the ratio of AUN is certainly A-gyo:UN-gyo \approx 3:97 even after repeated subculturing (**Supplementary Figure S1**). We also confirmed that the ratios get naturally back to be the original ratio of AUN (A-gyo:UN-gyo \approx 3:97) 7 days after culturing even when the ratios were intentionally prepared as A-gyo:UN-gyo = 10:90, 50:50, or 97:3 in advance (**Supplementary Figure S2**). These results potentially indicate that A-gyo has a symbiotic relationship (commensalism) with UN-gyo due to the Cys metabolism in A-gyo cell, as we previously reported.²⁷ Meanwhile, the artificial mixed ratios of AUN (A-gyo:UN-gyo = 15:85, 25:75, 50:50, or 97:3) caused Colon26-bearing BALB/c mice (N = 5 for each ratio) to die within 48 h. Therefore, we believe that nature-prepared “golden” ratio of AUN (A-gyo:UN-gyo \approx 3:97) has both of strong anticancer efficacy and high safety.”

Comment 2: Mechanisms of Thrombosis

The discussion should explore the multiple potential causes of AUN-induced thrombosis in more depth, including vascular invasion, toxin- and enzyme-induced damage, altered iron metabolism, and platelet-independent fibrinogen activation. Understanding the interplay of these mechanisms with tumor-specific thrombosis is essential.

Our Response 2: We have investigated the experiments suggested by the reviewer and concluded the mechanism as shown in **Figure 4k**. Here are the step-by-step answers.

(i) Vascular invasion

In order to clarify intratumor vascular invasion of AUN, we have performed NIR-II fluorescent imaging of the intratumor distribution of IR-1061-modified AUN using an NIR-II bioimager (please see **Supplementary Figure S41**). The following description has just been added in the main text.

Page 13, Line 12; “NIR-II fluorescent imaging⁷³ of IR-1061-modified AUN also demonstrated that the AUN was certainly distributed in a tumor over time in accordance with the tumor discoloration due to such bacterial destruction of intratumor vascular networks (**Supplementary Figure S41**).”

(ii) Toxin- and enzyme-induced damage

To investigate whether bacterial secreted toxins and enzymes have anticancer efficacy, we have performed cytotoxicity test and observation of spheroid destruction by using homogenized AUN (dead cell) (please see **Supplementary Figure S33**). The following description has also been added in the main text.

Page 11, Line 21; “To investigate whether these bacterial secreted toxins and enzymes have anticancer efficacy, we have performed cytotoxicity test and observation of spheroid destruction behavior by using homogenized AUN (dead cell) (**Supplementary Figure S33**). As we expected, the homogenized AUN displayed strong anticancer and oncolytic activities because of cytotoxic toxins and enzymes.”

(iii) Altered iron metabolism

We have surveyed iron metabolism in the tumor microenvironment by IHC staining and qPCR analyses of transferrin and ferritin (please see **Supplementary Figure S42**). The following description has just been added in the main text to support the hypothesis.

Page 13, Line 15; “Meanwhile, as we mentioned earlier, iron is essential for various cellular functions of cancer, and iron depletion strategies have a strong anti-proliferative effect on tumor cells.^{32–34} We found that transferrin and ferritin in the tumor microenvironment certainly were depleted by AUN treatment presumably owing to the increase of iron requirements of AUN via bacterial cross-talking (as indicated by the transcriptome analysis) (**Supplementary Figure S42**).”

(iv) Platelet-independent fibrinogen activation

We have confirmed that the tumor-specific discoloration is mainly caused by hematoma after i.v. administration of AUN with dosing various antiplatelet drugs or the combination of the mixture of antiplatelet drugs and heparin (**Supplementary Figure S31**). We have also performed coagulation test using all bacterial strains (AUN, A-gyo, and UN-gyo) in addition to homogenized AUN (dead cell) to make sure whether the proposed bacteria have coagulation ability or not. As the results, they do not show any coagulations at all because the bacteria do not secrete procoagulant molecules (e.g. thrombin-like enzymes) (please see **Supplementary Figure S38**). Therefore, the related descriptions have properly been amended as follows.

Page 1, Line 9; “Moreover, AUN induced an elegant mechanism based on unique bacterial features and intravascular biological reactions, such as intratumor vascular destruction, oncometabolites-triggered transformation of bacteria, and intratumoral bacterial colonization, and hence highly achieves tumor-targeted oncolysis, which work synergistically to achieve dramatic anticancer activity.”

Page 11, Line 4; “The AUN still caused tumor-specific discoloration and strong anticancer efficacy even after dosing various antiplatelet drugs or the combination of the mixture of antiplatelet drugs and heparin (**Supplementary Figure S30**). We confirmed that the tumor-specific discoloration was caused by hematoma not thrombi (**Supplementary Figure S31**). These results clearly indicate that PLT does not strongly affect antitumor efficacy but thrombus formation.”

Page 13, Line 20; “In sum, we conclude that the transcendent anticancer efficacy of AUN is mainly caused by tumor-specific vascular destruction thanks to the unique bacterial features such as transformation and hemolytic actions through an intratumoral bacterial population shift (**Figure 4k**). The selective destruction of tumor cells by the cytolytic bacterial exotoxins could strongly influence on the tumor suppression. The exhaustion of iron in the tumor milieu by the increased iron requirements of AUN also is involved in the tumor suppression.”

In addition, the following description has just been added in the main text.

Page 12, Line 35; “Besides, all bacterial strains themselves in addition to the homogenized AUN (dead cell)

do not have any coagulation properties at all because they do not secrete procoagulant molecules (e.g. coagulase and thrombin-like enzymes^{70,71}) (**Supplementary Figure S38**).”

Comment 3: Hemolysis Specificity

Clarification is needed on whether AUN-induced hemolysis is confined to tumor vasculature or affects normal red blood cells. Data suggesting possible effects on normal vasculature (e.g., **Figure 4i**) raise safety concerns for human application.

Our Response 3: We firstly apologize for causing misunderstand of the reviewer. Indeed, discoloration of the agar plate by AUN in the last picture just was deterioration of the ingredients in the plate. In fact, while A-gyo and UN-gyo exhibited β -hemolysis, AUN did not show any hemolysis at all on blood agar plates by careful double checking. The picture is replaced with the correct one. Besides, to further make sure the non-hemolysis behavior of AUN, we have studied hemolysis test using human blood cells (please see **Supplementary Figure S37**). As the results, AUN did not show hemolysis at all at any concentrations. The following description has been added in the main text.

Page 12, Line 32; “We also confirmed that hemolysis was not observed for human blood cells after treatment with AUN at any concentrations (**Supplementary Figure S37**). This result indicates that hemolytic A-gyo and UN-gyo might be attenuated toxicity each other when they are coexisted as AUN.”

Comment 4: Cytokine Effects and Tumor Necrosis

Further insights into the roles of TNF- α , IFN- γ , and other cytokines in vascular damage and necrosis across tumor models would strengthen the mechanistic framework. Comparing cytokine distribution and heterogeneity in tumor tissues could illuminate differences in therapeutic response.

Our Response 4: In order to address the comment from the reviewer, we have performed IHC staining analyses of cytokines including TNF- α , IFN- γ , IL-6, and IL-1 β in each tumor after i.v. single-dose administration of AUN. The following description has just been added in the main text.

Page 13, Line 34; “We consider that these differences of the speed of antitumor responses against each cancer type including hemorrhagic Colon26 could be related to the intratumoral expressions of inflammatory cytokines (**Supplementary Figure S46**). For instance, inflammatory cytokines in the highly responsible Colon26 were strongly expressed rather than the gradual responsive BxPC3. In any case, bacterial consortium AUN exhibits transcendent oncolytic ability against various murine- and human-derived malignant tumors in immunocompromised models.”

Comment 5: Translatability and Safety

The manuscript underestimates potential human risks due to physiological differences between mice and humans. Mice lacking baseline atherosclerosis and less prone to thrombosis may not adequately model human vascular risks. Incorporating models that better mimic human pathology, such as hyperlipidemic or

atherosclerosis-prone models, is essential for assessing safety.

Our Response 5: We have confirmed that AUN was totally safe by using lower-limb ischemia mice as an atherosclerosis-prone model. (**Supplementary Figure S24 and Supplementary Table S6**). Body weight measurement, blood test, and histological examination indicated that AUN has high biocompatibility against the lower-limb ischemia model mouse as well. The following description has just been added in the main text.

Page 9, Line 7; “We further confirmed that AUN is safe by using lower-limb ischemia mice as an atherosclerosis-prone model. (**Supplementary Figure S24 and Supplementary Table S6**).”

Comment 6: General Observations

Differences in tumor responses (e.g., hemorrhagic necrosis in Colon26 vs. gradual damage in BxPC3) suggest variability that warrants further exploration.

Our Response 6: As we already addressed in **Our Response 4**, to identify the differences in tumor responses related to hemorrhagic necrosis in Colon26 vs. gradual damage in BxPC3, we have carefully performed IHC staining analyses of inflammatory cytokines in Colon26 and BxPC3 tumors. The following description has been added in the main text.

Page 13, Line 34; “We consider that these differences of the speed of antitumor responses against each cancer type including hemorrhagic Colon26 could be related to the intratumoral expressions of inflammatory cytokines (**Supplementary Figure S46**). For instance, inflammatory cytokines in the highly responsible Colon26 were strongly expressed rather than the gradual responsive BxPC3. In any case, bacterial consortium AUN exhibits transcendent oncolytic ability against various murine- and human-derived malignant tumors in immunocompromised models.”

Comment 7: Figure 3c: TNF- α and fibrinogen staining require more interpretation, particularly fibrinogen’s biological significance in thrombosis and inflammation. Regarding thrombosis, fibrin should be examined rather than fibrinogen.

Our Response 7: We appreciate again for the reviewer’s beneficial comment. The image and analysis of IHC staining for fibrinogen has been replaced with the data of fibrin (Please see the revised **Figure 3c** and **3d**).

Comment 8: Figure 4j: The dynamics presented are unclear; a video-based explanation may enhance clarity.

Our Response 8: We have just added the videos according to the comment (please watch **Supplementary Videos S3–S5**).

Comment 9: Figure 5: The distinction between PBS and non-treatment controls needs a more precise definition.

Our Response 9: We have revised the applicable parts and the following description has also been added for the distinction between PBS and non-treatment controls.

Page 14, Line 1; “PBS was used as the placebo control against AUN. Whereas, the non-treatment group also was tested as another negative control without BxPC3 tumor inoculation and any administration of samples.”

Comment 10: Supplementary **Figure S2:** The bar graph colors should match the legend order.

Our Response 10: We apologize for the inconvenience. The bar colors (locations) have properly been amended. Many thanks again for your valuable comments and understanding.

Reviewer #3

Tumor-resident bacteria are a highly promising form of cancer therapy. This manuscript designed a bacterial consortium AUN composed of *Proteus mirabilis* and *Rhodopseudomonas palustris* and investigated the microbial characteristics and potential antitumor efficacy of AUN. The AUN demonstrated tumor-suppressive effects across multiple immunodeficient models. Furthermore, through integrated multi-dimensional analysis combining genomics, transcriptomics, histopathology, and *in vivo* imaging techniques, the study revealed AUN's tumor-targeting mechanisms mediated by thrombosis induction, bacterial oncolysis, and unique metabolic regulation. In conclusion, the design of the manuscript is innovative and offers a potential new strategy for treating tumors in immunosuppressed patients, but several key issues still need to be clarified.

General response: We are grateful to the reviewer for the dedicated assessment of our work, a compliment on the content, and encouragement. According to beneficial suggestions by the referee, we have carefully amended the manuscript.

Comment 1: The manuscript mentions that high-dose AUN administration caused mortality in mice, although this issue was mitigated through fractionated dosing. However, this observation suggests a narrow therapeutic window for AUN, necessitating cautious clinical implementation. It is recommended to explore either lower effective doses or combination therapy strategies to optimize safety and efficacy.

Our Response 1: We have investigated lower effective doses by verifying concentration of AUN (please see **Supplementary Figure S5**). The applicable parts in the main text have also been revised properly.

Comment 2: The manuscript's biosafety data focus on short-term side effects but lack long-term observations. The long-term safety of AUN and potential drug resistance issues require in-depth discussion.

Our Response 2: The long-term safety tests such as survival test and H&E staining of vital organs have been performed (please see the revised **Figure 2e and and Supplementary Figure S23**). The following description

has also been added in the main text.

Page 9, Line 5; “In addition, AUN did not show any toxicity in tissues 150 days after i.v. injection (**Supplementary Figure S23**). Indeed, hematoxylin and eosin (H&E) staining analyses demonstrated that the tissues of post i.v. injection of AUN resemble that of control group (PBS buffer).”

For addressing the issue about potential drug resistance of bacteria, we have performed the antibiotics-mediated controllability of AUN (**Supplementary Figure S4**), and added the following description in the main text.

Page 5, Line 6; “In fact, AUN could completely be eliminated from the mice by administration of antibiotics imipenem (**Supplementary Figure S4**). Despite such high controllability of AUN by antibiotics, concerns like potential drug resistance and infection risks associated with the use of live bacteria in treatments need careful consideration and resolution. The preliminary findings of the research, although promising, continued and expanded research are underscores to fully comprehend and further validate the potential risks between toxicity and efficacy.”

Comment 3: Although the mechanisms section is detailed, some conclusions may require more direct experimental evidence. For example, the manuscript mentions, "The exhaustion of iron in the tumor milieu by the increased iron requirements of AUN via bacterial cross-talking (as indicated by the transcriptome analysis) might also indirectly be involved in the tumor suppression." but there are no data to demonstrate the relationship between iron deprivation and tumor suppression directly.

Our Response 3: We have surveyed iron metabolism in the tumor microenvironment by IHC staining and qPCR analyses of transferrin and ferritin (please see **Supplementary Figure S42**). The following description has just been added in the main text to support the hypothesis.

Page 13, Line 15; “Meanwhile, as we mentioned earlier, iron is essential for various cellular functions of cancer, and iron depletion strategies have a strong anti-proliferative effect on tumor cells.³²⁻³⁴ We found that transferrin and ferritin in the tumor microenvironment certainly were depleted by AUN treatment presumably owing to the increase of iron requirements of AUN via bacterial cross-talking (as indicated by the transcriptome analysis) (**Supplementary Figure S42**).”

Comment 4: It was proposed in the manuscript that AUN induces thrombosis through activation of platelets (PLT) and fibrinogen, but experiments showed that antiplatelet drugs failed to inhibit their effects (**Figure S21**). This suggests that thrombosis may depend on other pathways (e.g., direct damage to the vascular endothelium by bacteria or release of procoagulant factors). Have specific procoagulant molecules secreted by AUN (e.g., thrombin-like enzymes) been detected? Is there evidence that AUN alters the coagulation status of the tumor microenvironment through metabolism, e.g., iron uptake?

Our Response 4: We have carefully confirmed that the tumor-specific discoloration is mainly caused by hematoma not thrombosis after i.v. administration of AUN with dosing various antiplatelet drugs or the combination of the mixture of antiplatelet drugs and heparin (**Supplementary Figure S31**). We have also performed coagulation test using all bacterial strains (AUN, A-gyo, and UN-gyo) in addition to homogenized AUN (dead cell) to make sure whether the proposed bacteria have coagulation ability or not. As the results, they do not show any coagulations at all because the bacteria do not secrete procoagulant molecules (e.g. thrombin-like enzymes) (please see **Supplementary Figure S38**). Therefore, the related descriptions have properly been amended as follows.

Page 1, Line 9; “Moreover, AUN induced an elegant mechanism based on unique bacterial features and intravascular biological reactions, such as intratumor vascular destruction, oncometabolites-triggered transformation of bacteria, and intratumoral bacterial colonization, and hence highly achieves tumor-targeted oncolysis, which work synergistically to achieve dramatic anticancer activity.”

Page 11, Line 4; “The AUN still caused tumor-specific discoloration and strong anticancer efficacy even after dosing various antiplatelet drugs or the combination of the mixture of antiplatelet drugs and heparin (**Supplementary Figure S30**). We confirmed that the tumor-specific discoloration was caused by hematoma not thrombi (**Supplementary Figure S31**). These results clearly indicate that PLT does not strongly affect antitumor efficacy but thrombus formation.”

Page 13, Line 20; “In sum, we conclude that the transcendent anticancer efficacy of AUN is mainly caused by tumor-specific vascular destruction thanks to the unique bacterial features such as transformation and hemolytic actions through an intratumoral bacterial population shift (**Figure 4k**). The selective destruction of tumor cells by the cytolytic bacterial exotoxins could strongly influence on the tumor suppression. The exhaustion of iron in the tumor milieu by the increased iron requirements of AUN also is involved in the tumor suppression.”

The following description has just been added in the main text.

Page 12, Line 35; “Besides, all bacterial strains themselves in addition to the homogenized AUN (dead cell) do not have any coagulation properties at all because they do not secrete procoagulant molecules (e.g. coagulase and thrombin-like enzymes^{70,71}) (**Supplementary Figure S38**).”

Next, we have surveyed iron metabolism in the tumor microenvironment by IHC staining and qPCR analyses of transferrin and ferritin to support the aforesaid hypothesis (please see **Supplementary Figure S42**). The following description has also been added in the main text.

Page 13, Line 15; “Meanwhile, as we mentioned earlier, iron is essential for various cellular functions of cancer, and iron depletion strategies have a strong anti-proliferative effect on tumor cells.³²⁻³⁴ We found that transferrin and ferritin in the tumor microenvironment certainly were depleted by AUN treatment presumably owing to the increase of iron requirements of AUN via bacterial cross-talking (as indicated by the

transcriptome analysis) (**Supplementary Figure S42**).”

Comment 5: A-gyo accounted for only 3% of AUN (97% of UN-gyo), but the gene expression profiles of the consortium were significantly different from those of the single strains (**Fig. 1e-f**). How do the two regulate thrombosis and tumor lysis through cross-talk? Are there co-culture experiments to demonstrate their synergistic effects?

Our Response 5: Firstly, in order to address the comment from the reviewer, we have compared cytotoxicity of all bacteria (UN-gyo, A-gyo, and AUN) against Colon26 cell line (please see **Supplementary Figure S34**). As the results, AUN apparently showed the highest cytotoxicity against Colon26 cancer cells owing to synergistic effects caused by the co-culturing UN-gyo with A-gyo (dynamic structural transformation of A-gyo). The following description has also been added in the main text.

Page 11, Line 31; “AUN showed the highest cytotoxicity against Colon26 in comparison with A-gyo and UN-gyo (**Supplementary Figure S34**).”

Subsequently, hemolysis and coagulation tests have also been done although we have already responded in **Our Responses 4 and 12** (please see **Supplementary Figures S37 and S38**). Although AUN does not exhibit any hemolytic and coagulation activity at all, when AUN reaches to a tumor, A-gyo in one of the components of AUN dramatically expresses cytolytic-induced tumor-specific vascular and cancer cell destructions because of oncometabolites-triggered transformation of A-gyo and intratumoral bacterial population shift (please see **Figure 4 and Supplementary Figures S36 and S39**). We have also surveyed iron metabolism in the tumor microenvironment by IHC staining and qPCR analyses of transferrin and ferritin (please see **Supplementary Figure S42**). As the result, transferrin and ferritin in the tumor microenvironment certainly were depleted by AUN treatment presumably owing to the increase of iron requirements of AUN via bacterial cross-talking, as indicated by the transcriptome analysis (please see **Figure 1f**). Therefore, we believe that these various unique bacterial features through cross-talking are important to express significant targeting angiolysis and oncolysis.

Comment 6: In the manuscript, the authors refer to “AUN are beyond species difference, being highly effective as a universal therapeutic agent in various immunocompromised models derived from both human and murine cancer cells.” However, validating the tumor effect of AUN in the manuscript focuses on colon and pancreatic cancer models. There is limited data on other tumor types, and the authors must correct this description.

Our Response 6: We have omitted the description to avoid misunderstanding of audiences.

Comment 7: BALB/c-nu/nu, SCID, and NOD-SCID models were used in the manuscript, but these models still retain some innate immunity (e.g. residual NK cells in NOD-SCID). Did the authors test the role of innate immune cells, such as neutrophils and macrophages, in thrombosis? In addition, patients with clinical

immunosuppression are often accompanied by complex immune dysregulation due to chemotherapy or radiotherapy. Consider whether the efficacy of AUN is stable in such scenarios.

Our Response 7: We truly appreciate again for the excellent insights of the reviewer. We have investigated the role of neutrophils whether they influence on the thrombosis and anticancer efficacy (please see **Supplementary Table S2 and Supplementary Figure S10**). The following description has also been added in the main text.

Page 7, Line 4; “In fact, the mice were not died at least one week even after administration of a fatal dose of AUN (15×10^9 CFU/mL) after depletion of neutrophils by using anti-Ly6G antibody (**Supplementary Table S2**). Moreover, depletion of neutrophils via anti-Ly6G antibody significantly improved tumor clearance even by a single low dose administration of AUN (3×10^9 CFU/mL) even with the administration of the combination of antiplatelet drugs (the mixture of cilostazol, acetylsalicylic acid, and ticlopidine hydrochloride) and anticoagulation heparin. (**Supplementary Figure S10**). These results indicate that neutrophils may impede the spread of bacteria through the tumor and prevent complete oncolysis.³⁹”

To address the issue raised by the reviewer about the immune dysregulation due to chemotherapy or radiotherapy, we have tested the dexamethasone-premedicated antitumor efficacy of AUN because steroid could often be used to control immune abnormalities with inflammation for immune dysregulated patients. As the results, AUN expressed stable high efficacy even in such anti-inflammatory and immunosuppressive conditions (please see **Supplementary Figure S19**). The following description has also been added in the main text.

Page 8, Line 28; “Notably, AUN expresses stable and high antitumor efficacy in such anti-inflammatory and immunosuppressive conditions caused by premedication of dexamethasone (**Supplementary Figure S19**).”

Comment 8: Immunodeficient mouse models do not fully mimic clinically immunosuppressed patients, and the authors need to discuss the possible risk of immune clearance of AUN in humans.

Our Response 8: We have added the following description in Discussion.

Page 14, Line 35; “Besides, although transient immune clearance was observed by administration of AUN, as evidenced by the results of various safety tests, health conditions of the treated mice return back to normal because of homeostatic strong immune resilience. However, we envision that long-term and careful assessments of immunobiological and physiological factors using large animals are further essential for future clinical trials.”

Comment 9: **Figure 3e** demonstrates only the H&E staining of the thrombus and can be supplemented with other characterization evidence to enhance the reliability of the conclusions.

Our Response 9: We have performed additional experiment to enhance the reliability of the conclusions (see **Supplementary Figures S10 and S28**). The following description has also been added in the main text.

Page 10, Line 5; “Tumor cells and blood vessels are intertwined in a complex manner, and thrombi are observed widely and randomly 6 h after the AUN administration (**Supplementary Figure S28**). The thrombi are mainly platelet aggregates, and neutrophils may be trapped. Blood vessels without thrombi are also present. We believe that neutrophils are not a significant factor for thrombi formation because tumor-specific thrombuses could be observed after AUN administration with the depletion of neutrophils via anti-Ly6G as we mentioned above (**Supplementary Figure S10**).”

Comment 10: **Figure 4** has a low resolution for **b** and **f**. The authors need to provide clearer optical microscopic images.

Our Response 10: We apologize for the inconvenience to evaluate the **Figure 4b** and **4f**. The panels have been replaced with higher resolutions.

Comment 11: Although AUN is described as “non-pathogenic”, A-gyo carries several hemolysin and toxin genes (**Figure 4h**). Has its long-term toxicity to normal tissues (e.g. liver, kidney) been assessed? Can the toxicity be further reduced by gene editing?

Our Response 11: As we already answered in **Our Response 2**, the long-term safety tests such as survival test and H&E staining of vital organs have been performed (please see the revised **Figure 2e** and **Supplementary Figure S23**).

Meanwhile, we really appreciate again for sharing the excellent idea to use gene editing to attenuate bacterial toxicity. However, the establishment of this technique requires long time for application of the current study. It is also fact that there are several drawbacks of gene-engineered bacteria especially due to the complicated legal regulations (limited usage issues and costs of equipment, facility, and shipping etc.) in addition to the establishment issue of the technique. Therefore, we honestly focus on “natural” bacteria toward future clinical trial. That said, to overcome the limitations of gene-engineering, we recently reported that a simple scaffold-mediated bacterial culture is useful for bacterial attenuation and enhancing the anticancer therapeutic abilities of AUN [Miyahara et al., *Chemical Engineering Journal* **499**, 156378 (2024)]. In any case, the following description including gene editing idea proposed by the reviewer has just been added in the Discussion.

Page 15, Line 4; “Whereas, gene engineering¹⁰⁻¹⁵ and our recent reported scaffold-mediated bacterial culturing method⁷⁴, those biomedical engineering approaches are useful for attenuation of toxicity, would be applicable to develop safe and secure bacterial modality. The potential capability of AUN for cancer treatment would be promising as a blockbuster drug.”

Comment 12: The authors need to provide data on the hemolytic activity of AUN in human cell lines to

support clinical safety.

Our Response 12: We firstly apologize for causing misunderstand of the reviewer. Indeed, discoloration of the blood agar plate by AUN in the last picture just was deterioration of the ingredients in the plate. In fact, while A-gyo and UN-gyo exhibited β -hemolysis, AUN did not show any hemolysis at all on blood agar plates by careful double checking. The picture has been replaced with the correct one. Besides, to further make sure the non-hemolysis behavior of AUN, we have studied hemolysis test using human blood cells (please see **Supplementary Figure S37**). As the results, AUN did not show hemolysis at all at any concentrations. The following description has been added in the main text. Many thanks again for your valuable comments and understanding.

Page 12, Line 32; “We also confirmed that hemolysis was not observed for human blood cells after treatment with AUN at any concentrations (**Supplementary Figure S37**). This result indicates that hemolytic A-gyo and UN-gyo might be attenuated toxicity each other when they are coexisted as AUN.”